# MCP-FLOW: FACILITATING LLM AGENTS TO MASTER REAL-WORLD, DIVERSE AND SCALING MCP TOOLS

## ABSTRACT

Large Language Models (LLMs) increasingly rely on external tools to perform complex, realistic tasks, yet their ability to utilize the rapidly expanding Model Contextual Protocol (MCP) ecosystem remains limited. Existing MCP research covers few servers, depends on costly manual curation, and lacks training support, hindering progress toward real-world deployment. To overcome these limitations, we introduce MCP-Flow, an automated web-agent-driven pipeline for large-scale server discovery, data synthesis, and model training. MCP-Flow collects and filters data from 1166 servers and 11536 tools, producing 68733 high-quality instruction-function call pairs and 6439 trajectories, far exceeding prior work in scale and diversity. Extensive experiments demonstrate MCP-Flow's effectiveness in driving superior MCP tool selection, function-call generation, and enhanced agentic task performance. MCP-Flow (available at *[URL]*.) thus provides a scalable foundation for advancing LLM agents' proficiency in real-world MCP environments.

## 1 INTRODUCTION

The development of tool-using agents has rapidly accelerated in recent years, driven by the need for Large Language Models (LLMs) to perform complex, realistic tasks beyond pure language generation (Shen, 2024; Yan et al., 2025). A key milestone in this area is the Model Contextual Protocol (MCP), a framework designed to enhance models' interactions with diverse external tools in a unified manner (Anthropic, 2024a). Unlike traditional APIs with fixed interfaces and limited flexibility, MCP offers a dynamic, context-aware environment, allowing LLM agents to adapt to heterogeneous servers and continuously evolving functionalities (Yin et al., 2025; Mo et al., 2025).

The integration of MCP presents both extensive opportunities and pressing challenges for tool-using agents, spurring a surge of research and benchmarks designed to evaluate model proficiency in MCP utilization. Empirical evidence (Luo et al., 2025b; Liu et al., 2025) shows that even state-of-the-art (SOTA) LLM agents struggle to fully exploit MCP tools, underscoring the need for more comprehensive frameworks and concerted efforts to improve their mastery of real-world MCP environments. The obstacle lies in the gap between the complexity, diversity and rapid proliferation of real-world MCP servers, and the limited ability of current LLMs to utilize them.

Addressing this gap requires not only improved evaluation but also training resources that expose models to realistic MCP scenarios. However, no large-scale, high-quality datasets are currently available to fulfill this critical requirement, and existing MCP studies have yet to undertaken the essential step towards data construction. As summarized in Table 1, they exhibit several notable limitations: (1) most of them rely on only a small number of servers ($\leq$ 20) and tools, resulting in limited coverage, diversity and scalability; (2) they predominantly depend on labor-intensive human data collection, which cannot keep pace with the rapid growth of open-source initiatives and the continual emergence of new MCP servers; and (3) no existing MCP frameworks offer training support, thereby failing to translate benchmark results into tangible improvements of LLMs.

To resolve the dataset absence, in this paper, we aim to introduce an approach for automatically constructing high-quality datasets from a large number of MCP servers, thereby facilitating the more effective utilization of real-world and continuously scaling MCP tools by LLM agents. That said, building such an automated training framework is non-trivial, as several challenges remain: (1) Each MCP server maintains its own repository and documentation. How to perform automated data acquisition on a large scale while supporting dynamic updates? (2) With thousands of server

configurations, how to ensure the unified generation of high-quality and diverse datasets? (3) How can the resulting datasets be effectively leveraged to strengthen MCP capability for both closed-ended and open-ended models, and even support complex agentic tasks?

To tackle these challenges, we propose **MCP-Flow**, a comprehensive pipeline, data and model suite that automatically constructs datasets from real-world, diverse and continuously scaling MCP servers, thereby facilitating more effective utilization of MCP tools by LLM agents. MCP-Flow comprises two key components: server discovery and data synthesis. Specifically, we first introduces a novel web-agent–based automatic collection pipeline. Building on the contributions of MCP marketplaces which aggregate a wealth of servers, our approach leverages web agents to automate server discovery and acquisition. This implementation simplifies adaptation to new platforms and require only incremental updates, thus accommodating the continuous emergence of real-world servers.

Second, we propose a scalable data synthesis pipeline, which comprises two main stages: data generation and data filtration. We adopt a few-shot generation approach grounded in tool information, therefore yield both instructions and corresponding ground-truth tool labels. To ensure instruction specificity and diversity, we incorporate slot-fill revision and WizardLM evolution, which populate each required parameter slot with valid values and rewrite instructions to increase difficulty and promote reasoning (Xu et al., 2023), respectively. After rigorous filtering based on multiple criteria, we obtain a dataset encompassing 1,166 servers and 11,536 tools, exceeding the scale of all previous work combined, with 356 servers producing valid responses and trajectories. In total, the resulting dataset contains 68733 instruction-function call pairs and 6,439 trajectories suitable for training.

Three versatile approaches demonstrate how MCP-Flow datasets empower LLM agents to maximize their engagement with MCP tools: (1) training LLMs (particularly small-size models) to significantly advance their real-world MCP tool-use capabilities; (2) establishing a retrieval database that can augment closed-source models in MCP tool usage; and (3) serving as a playground for evaluating MCP servers and tools themselves, rather than focusing solely on model performance.

We conduct extensive experiments on data from six MCP marketplaces, comparing MCP-Flow with 10+ strong LLMs and latest baselines. The models are evaluated on three test splits covering both seen and unseen scenarios, and further assessed on the standard agentic benchmark GAIA (Mialon et al., 2023). The results demonstrate that: (1) Contemporary SOTA models still exhibit suboptimal performance on real-world MCP tool utilization, and their effectiveness further deteriorates as the number of candidate tools increases. (2) Our trained MCP-Flow models consistently outperform SOTA models in both MCP tool selection and function call formatting, despite substantially smaller size. (3) By retrieving examples from MCP-Flow, closed-ended models such as GPT-4o (OpenAI, 2024) can further enhance their performance in MCP tool utilization. (4) By replacing initial tool invocation, MCP-Flow can even improve agent performance on multi-turn, complex tasks while simultaneously reducing inference costs. (5) Collected MCP servers exhibit diverse characteristics and varying quality for identical tasks; MCP-Flow lays the groundwork for future research to systematically compare MCP servers and tools. In summary, our contributions are as follows:

1. We propose MCP-Flow, a web-agent-driven automated pipeline capable of constructing datasets that accommodate real-world, diverse, and continuously scaling MCP servers.
2. We release a large-scale, high-quality dataset that supports LLM training, evaluation and retrieval augmentation, thereby enhancing both closed-ended and open-ended models on MCP utilization.
3. We offer a compact, fine-tuned LLM series; the extensive experiments verify the value of MCP-Flow in both MCP tool selection and formatting, and the facilitation of agentic tasks.

## 2 RELATED WORK

### 2.1 TRADITIONAL API-BASED DATASETS AND BENCHMARKS

The integration of LLMs with external tools has emerged as a critical research direction for extending model capabilities beyond their inherent knowledge. The data for tool learning are collected either through automated synthetic data generation or relying on real-world APIs. Synthetic data generation approaches including APIGen (Liu et al., 2024b) and ToolACE (Liu et al., 2024a) address data scarcity through automated frameworks. However, they suffer from fundamental concerns regarding real-world applicability, as artificially constructed interactions may not capture authentic tool usage complexity and error patterns encountered in production environments. ToolLLM (Qin et al., 2023) significantly expanded scope by incorporating real-world REST APIs from RapidAPI Hub,

Table 1: Comparison of representative datasets and benchmarks for LLM tool usage. Compared to contemporary MCP studies, MCP-Flow covers a substantially larger number of MCP servers, provides an automated pipeline for newly uploaded servers, and supports model training.

| Dataset/Benchmark | Tool Type | | Training Support | Data Scale (N)# | | | |
| Chronological order | Real-World | Scalable | | Source | Server | Tool/API | Sample |
|---|---|---|---|---|---|---|---|
| **Traditional API-Based** | | | | | | | |
| ToolBench (Qin et al., 2023) | ✓ | ✗ | ✓ | 1 | – | 3451 | 12.6k |
| $\tau$-Bench (Yao et al., 2024) | ✓ | ✗ | ✗ | 2 | – | < 30 | – |
| APIGen(xALM) (Liu et al., 2024b) | ✗ | ✓ | ✓ | 1 | – | 3673 | – |
| ToolACE (Liu et al., 2024a) | ✗ | ✓ | ✓ | 1 | – | – | 11.3k |
| **Modern MCP-Specialized** | | | | | | | |
| MCPBench (Luo et al., 2025a) | ✓ | ✗ | ✗ | 1 | 10 | 10 | – |
| MCP-Zero (Fei et al., 2025) | ✓ | ✗ | ✗ | 1 | 308 | 2797 | 0 |
| LiveMCPBench (Mo et al., 2025) | ✓ | ✗ | ✗ | 1 | 70 | 527 | 95 |
| MCPToolBench++ (Fan et al., 2025) | ✓ | ✗ | ✗ | 3 | 12 | 87 | 1509 |
| MCP-Universe (Luo et al., 2025b) | ✓ | ✗ | ✗ | 1 | 11 | 133 | 231 |
| MCP-Flow (Ours) | ✓ | ✓ | ✓ | **7** | **1166** | **11536** | **68733** |

introducing sophisticated planning algorithms through depth-first search-based approaches. Despite notable progress in API-based datasets, current studies still face critical limitations, as REST APIs are often unstable and lack standardized protocols. With the flourishing of MCP, research efforts increasingly focus on unified and reliable MCP tools rather than traditional APIs, aiming to bridge the gap between existing datasets and practical MCP scenarios.

## 2.2 MCP-Specialized Datasets and Benchmarks

With its increasing maturity and widespread adoption, MCP now offers extensive opportunities for model training, evaluation, and real-world deployment (Hasan et al., 2025). While the proliferation and success of MCP have stimulated considerable research interest, current approaches still encounter three key limitations: (1) Most studies still rely on manually curated collections of MCP servers and tools. For instance, MCPToolBench++ (Fan et al., 2025) reports 4,000 servers but experiments on only 12. Similarly, MCP-Zero (Fei et al., 2025) pioneered MCP tool discovery with 308 servers drawn from the official MCP website, but it operates exclusively on a simple position-based retrieval task, without user queries or instruction-following contexts from any of these servers. (2) Current server collection and data construction depend heavily on human efforts (Luo et al., 2025a; Mo et al., 2025) or human-curated crawling code (Lin et al., 2025). No automated pipeline yet exists to keep pace with the rapidly evolving number of MCP servers released in the community. (3) Despite the revelation of performance limitations in even SOTA models, current benchmarks (Liu et al., 2025; Luo et al., 2025b) function solely as evaluation platforms, failing to address the critical challenge of training data scarcity that fundamentally constrains the effectiveness of LLMs in MCP environments.

This gap is particularly concerning given the critical importance of realistic training datasets for developing tool-augmented systems capable of utilizing thousands of available MCP tools. The absence of systematic data collection pipelines and large-scale MCP-specific datasets poses a major barrier to fully realizing the potential of the unified MCP ecosystem. In contrast to previous works, we propose an agent-based pipeline which automates the process of collecting newly updated servers and building a high-quality dataset featuring real-world, diverse, and scaling MCP servers.

## 3 MCP-Flow

In this section, we introduce MCP-Flow's automated data construction pipeline, which begins with web-agent–based server and tool collection (Section 3.1), followed by scalable data synthesis comprising two stages: data generation (Section 3.2) and data filtration (Section 3.3). We also provide statistical analyses of sample counts in Table 2; data distributions and diversity in Figure 3; and a representative example illustrating all components of the constructed dataset in Figure 4.

### 3.1 Automated MCP Server & Tool Collection

In the first stage, we collect a large set of MCP servers from diverse sources and endpoints. The scale of our collection surpasses the combined size of all existing MCP-related studies. After deduplication, we then collect tool information through local deployment.

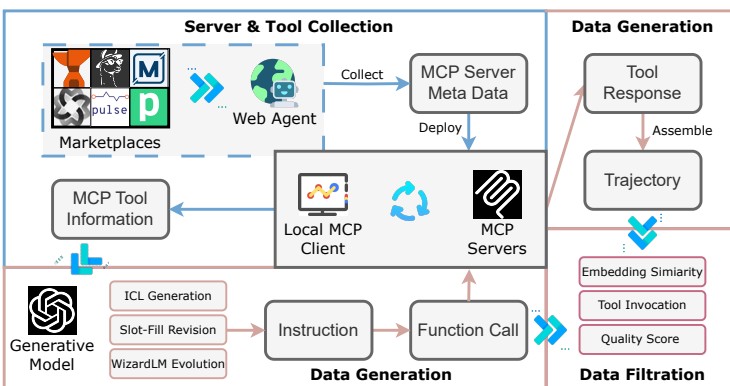

Table 2: Dataset statistics of servers, tools, function calls and trajectories.

| Type | Count |
| --- | --- |
| Server | 1166 |
| Tool | 11536 |
| Train | 52169 |
| Seen Test | 5216 |
| Unseen Tool | 5249 |
| Unseen Server | 6099 |
| Trajectory | 6439 |

Figure 1: Pipeline overview. MCP-Flow initiates with automated server discovery from various marketplaces, and proceeds through scalable data synthesis (diverse generation + rigorous filtering).

**Web-Agent Automated Server Crawling.** To accommodate the rapid growth of MCP servers, we propose an automated collection pipeline primarily driven by web agents. Specifically, we employ Playwright, an MCP-compatible web agent, to systematically navigate widely used MCP marketplaces and websites. This implementation ensures timely adaptation to modifications and newly added servers. The process is illustrated in Figure 2, with efficiency analysis in Section B.5.

Within a human-defined workflow, the agent autonomously navigates to the target server's dedicated page and retrieves its configuration file (in JSON format) via page snapshots. Our pipeline supports various platforms, Smithery, Glama, MCP.so, MCPHub, PipeDream, and PulseMCP (DeepNLP). In principle, this web agent–based approach is applicable to any well-structured website with minimal human modification. Unlike traditional web crawlers, which typically require detailed parsing logic tailored to each website's HTML structure, our approach operates with high-level instructions, avoiding low-level code dependencies. This design not only offers cross-platform generalization but also simplifies adaptation to future marketplaces and improves operational flexibility. Importantly, after our large-scale crawling and pre-processing, researchers only need to execute the pipeline incrementally for newly released servers, rather than restarting the entire process. This design significantly minimizes both time and computational costs.

**Server Deduplication.** Some popular servers may appear on multiple websites. For example, *context7* is included on all supported endpoints. It is therefore necessary to perform deduplication, even when the servers are listed under different names, interfaces and configurations. After an in-depth examination of inter-server differences, we conclude that the most reliable criterion for distinguishing servers is not their names or providers, but rather the tool descriptions. If two servers share an identical list of tool descriptions, we treat them as the same entity.

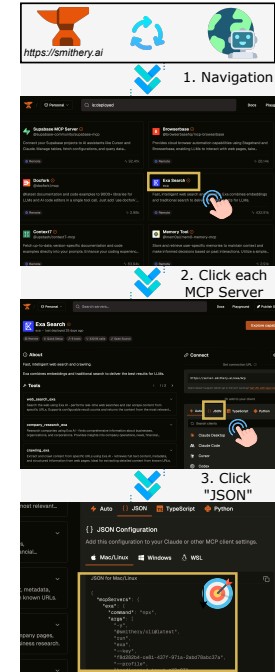

Figure 2: Process of web-agent automated server crawling with more details in Appendix D.

**Local Deployment and Tool Collection.** Using the collected configurations, we deploy servers locally via our MCP client. For standard input/output (stdio)–based servers, deployment is handled through npm and uvx, while servers using Server-Sent Events (SSE) are connected via their URLs. The client implementation builds upon the popular repository dolphin-mcp. For each successfully deployed server, we extract its tool information, including tool name, description, and parameters (input schema). Certain servers require API keys or access to specific software. Due to their sensitivity, these servers cannot be automatically deployed and are therefore excluded. Nevertheless, we highlight the investigation of such personalized servers as an interesting direction for future work, as discussed in Appendix A.

### 3.2 INSTRUCTION GENERATION AND TRAJECTORY COLLECTION

In the second stage, we generate diverse instructions with ground-truth function calls, and collect tool responses to form the trajectories. Details are provided in Appendix D with prompts in Appendix E.1.

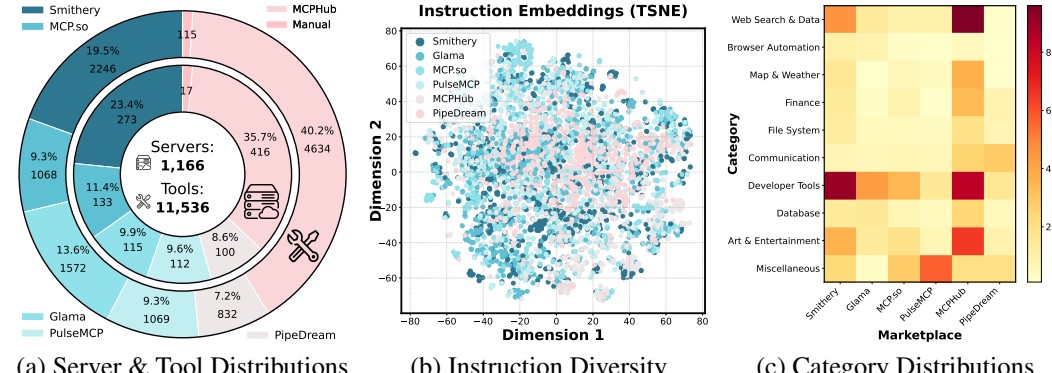

(a) Server & Tool Distributions      (b) Instruction Diversity      (c) Category Distributions

Figure 3: Dataset statistics. (a) MCP-Flow encompasses a large-scale collection of MCP servers and tools from six distinct marketplaces. (b) T-SNE visualization of instruction embeddings (1,000 random samples per marketplace) shows the dataset diversity. (c) Each MCP server is classified into one of ten categories, with the distributions reflecting the heterogeneity across marketplaces.

**Few-Shot Generation.** To construct datasets with inherent ground-truth labels, we initiate generation from the tool perspective. Our focus is on diversity; therefore, for each tool, the model generates five distinct instructions based on human-curated examples, all of which require the use of the target tool.

**Slot-Fill Revision.** Next, to enhance instruction detail and ensure that all tool-required input parameters are specified, we employ a slot filling process. Each parameter required by the tool is treated as a slot. If a slot is not provided in the original query, a valid value is automatically generated. The query is then revised to include these new parameters for fluency. After slot filling, some instructions may still contain placeholders. We then apply specialized regular-expression rules (§D.3) to detect such strings and replace them with pre-collected candidates from our local system.

**WizardLM Evolution.** To further improve instruction complexity and diversity, we adopt the evolution method proposed in Xu et al. (2023). The process randomly selects an evolution direction for each query, such as concretization or reasoning. We set the evolution depth to 2 to balance generation cost and output quality. After evolution, we obtain a large scale tool-specified instruction sets with high diversity and complexity. The instructions are then filtered as described in Section 3.3.

**Function Call Generation.** Given the ground-truth tool, its input schema, and the corresponding instruction, we prompt GPT-4o to generate the formalized function call. This generation is straightforward for strong models such as GPT-4o, as it involves no interference factors. With additional filtration in Section 3.3, the correctness of the generated function calls can be substantially ensured.

**Tool Response Collection.** Based on the function calls, we collect tool responses by communicating with the MCP server via the local client. The assistant model then summarizes these outcomes and generates the final response. By integrating all components, we obtain the complete trajectory.

### 3.3 RIGOROUS DATA FILTRATION

**Embedding Similarity Filtration.** We filter instructions using a combination of rule-based and LLM-based measurements. In particular, some original instructions are overly similar to the tool descriptions, which is undesirable since such queries make tool selection trivial. To address this, we compute the instruction–description embedding similarity. We empirically set a threshold of 0.8 and discard instructions that exceed this value. Embedding details are provided in Section B.2.

**Tool Invocation Filtration.** We further ensure the reliability of ground-truth tool labels by instructing GPT-4o and DeepSeek-V3 to select the correct tool from the labeled tool and two randomly sampled candidates. Instructions for which both models fail to identify the expected tool under this simplified setting are discarded. Negligible instruction removals confirming the tool annotation reliability.

**Quality Score Filtration.** To evaluate the quality of instructions and function calls, we employ DeepSeek-V3, a SOTA reasoning model different from GPT-4o we use for generation. As previously indicated by LiveMCPBench, DeepSeek-V3 (DeepSeek-AI, 2024) achieves strong agreement with human annotators, thus serving as a good judge model. We discard instructions and function calls that receive a quality score below a threshold of 6/10. Evaluation prompts are provided in Appendix E.1.

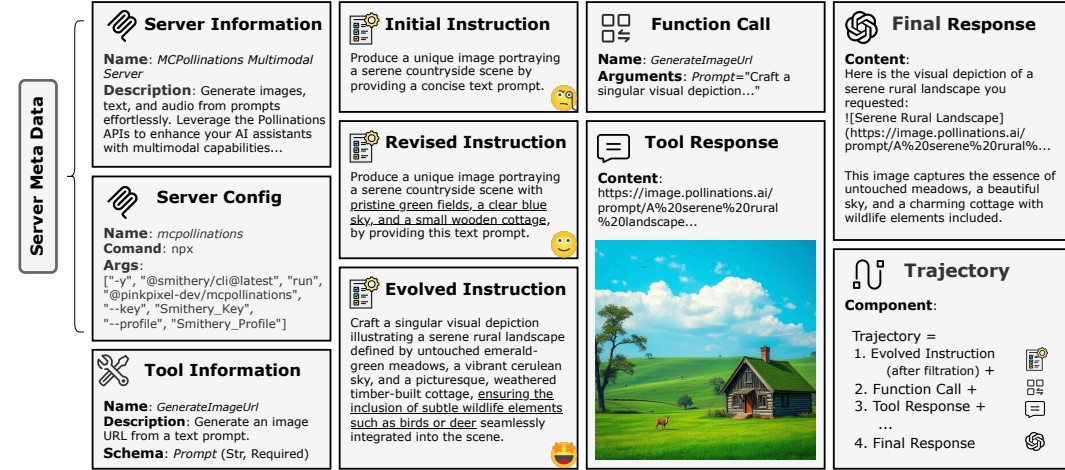

Figure 4: Data example using the *MCPollinations Multimodal Server* from `Smithery`. The first column is collected as described in Section 3.1, and the remaining data as described in Section 3.2. The server returns a URL linking to the image shown above. Note that all 1,166 servers have corresponding tool information and generated function calls, but not all yield valid tool responses.

**Trajectory Filtration.** Trajectories are more vulnerable to inconsistencies, as they require valid responses from external tool providers. In practice, many servers demand specific setups (e.g., API keys, personal workspaces, or software dependencies), while others may be temporarily unavailable even though their metadata remain valid. We filter out trajectories with invalid tool responses collected under such conditions, also using DeepSeek-V3 as a judge.

## 4 EXPERIMENTS

In this section, we evaluate the fine-tuned models from the MCP-Flow suite in terms of tool selection and formatting capabilities (Section 4.2). We further assess the effectiveness of MCP-Flow in enhancing non-trainable models for both function-call generation (Section 4.3) and complex agentic tasks (Section 4.4). Ablation studies are presented in Section 4.5. Supplementary experiments, including server evaluation, are attached in Appendix B.

### 4.1 BASIC SETUPS

**Models and Baselines.** To demonstrate the utility of our datasets, we compare various SOTA models and the latest MCP datasets. The model suite includes close-ended models including GPT-4o, Claude-4-Sonnet, open-ended reasoning models like Qwen3-8B (Team, 2025), Llama3.1-8B (Meta, 2024), tool-specialized models like ToolACE-8B (Liu et al., 2024a). We also compare between datasets by finetuning models on our datasets and others like MCPToolBench++ (Fan et al., 2025).

**Data Splits.** To ensure a fair comparison and avoid data contamination under both seen and unseen scenarios, we divide our full dataset into four data splits. For each marketplace, we first randomly split the servers with a 12:1 ratio, assigning the held-out portion as the *unseen-server* subset. Within the remaining (seen) servers, we further split all tools belonging to these servers at an 11:1 ratio, assigning the held-out portion as the *unseen-tool* subset. The remaining samples are then divided into a training set and a *seen-test* subset with a 10:1 ratio. The training split and the three test splits are in a 10:1:1:1 ratio in expectation. In total, this process yields 6 marketplaces × 4 splits = 24 subsets.

**Training Configuration.** We employ LoRA finetuning (Hu et al., 2022) based on the implementation of LLaMA-Factory (Zheng et al., 2024). In most experiments we set the training tool size to 10 and randomly sample candidate tools from the seen tool pool to form the training set for each instruction-function call pair. If not otherwise mentioned, we test models after training for one epoch to prevent over-fitting. Key parameters and environment details are provided in Appendix C.1.

**Metrics.** Following prior works, we report both rule-based metrics and LLM-as-a-judge metrics. Specifically, to evaluate models' tool use capability to generate function calls, we compute tool accuracy (denoted as *Tool*) (Fei et al., 2025; Yuan et al., 2024), which measures the model's tool-selection capability; as well as parameter accuracy (denoted as *Param*) (Han et al., 2025); and abstract

Table 3: Comparison of MCP tool selection and formatting capabilities across various models using 10 tools. MCP-Flow models achieve the best performance notably notably small model sizes, while SOTA LLMs (e.g., Claude-4-Sonnet) exhibit suboptimal performance even under this simple setting.

| Category | Backbone Model | Seen Test | | | Unseen Tool | | | Unseen Server | | |
|---|---|---|---|---|---|---|---|---|---|---|
| | | Tool | Param | AST | Tool | Param | AST | Tool | Param | AST |
| **Large Models (> 10B) through Azure API** | | | | | | | | | | |
| Closed-Ended | GPT-4o | 88.6 | 68.2 | 58.8 | 85.0 | 71.4 | 62.0 | 81.4 | 55.6 | 50.8 |
| | GPT-4.1 | 77.6 | 61.4 | 52.2 | 71.8 | 62.8 | 54.8 | 70.8 | 51.4 | 45.4 |
| | Claude-4-Sonnet | 85.8 | 68.6 | 56.6 | 83.0 | 74.4 | 63.6 | 72.6 | 56.0 | 48.4 |
| | Gemini-2.5-Pro | 54.2 | 42.8 | 36.8 | 55.4 | 50.6 | 42.4 | 49.6 | 38.0 | 32.2 |
| Open-Ended | DeepSeek-V3 | 84.2 | 67.2 | 58.8 | 82.0 | 70.8 | 59.8 | 77.6 | 55.6 | 48.8 |
| | Kimi-K2 | 85.6 | 68.0 | 55.0 | 82.4 | 73.0 | 60.2 | 74.4 | 53.0 | 47.4 |
| **Small Models (≤ 10B) through Local Deployment** | | | | | | | | | | |
| General Model with Reasoning | Qwen3-0.6B | 59.2 | 44.6 | 35.4 | 62.6 | 46.6 | 34.0 | 59.2 | 45.6 | 38.2 |
| | Qwen3-4B | 79.6 | 66.2 | 57.8 | 81.8 | 75.4 | 65.4 | 74.8 | 55.2 | 46.8 |
| | Qwen3-8B | 83.6 | 69.6 | 59.6 | 86.0 | 76.4 | 65.6 | 76.4 | 57.2 | 48.2 |
| | Llama3.1-8B | 75.8 | 55.0 | 32.4 | 74.0 | 53.6 | 33.8 | 74.8 | 55.8 | 33.2 |
| Tool-Specialized | Groq-8B-Tool-Use | 39.4 | 22.8 | 20.6 | 45.6 | 26.2 | 23.6 | 42.6 | 22.8 | 15.8 |
| | ToolACE-8B | 89.4 | 49.8 | 45.6 | 83.4 | 49.0 | 44.4 | 88.4 | 51.8 | 46.0 |
| MCPToolBench++ | Qwen3-0.6B | 76.4 | 57.0 | 47.4 | 80.2 | 57.2 | 47.0 | 70.0 | 46.4 | 35.8 |
| | Qwen3-4B | 91.4 | 77.2 | 62.2 | 91.6 | 76.0 | 63.0 | 80.4 | 57.2 | 47.6 |
| MCP-Flow (Ours) | Qwen3-0.6B | 96.8 | 87.2 | 75.4 | 98.2 | 86.8 | 75.2 | 98.4 | 70.6 | 58.0 |
| | Qwen3-4B | **99.2** | **91.8** | 81.2 | 98.6 | **91.4** | **78.2** | 98.4 | 72.2 | 59.8 |
| | Llama3.1-8B | 98.6 | 91.0 | **81.6** | **99.0** | 91.2 | 77.6 | **99.4** | **77.0** | **65.2** |

Table 4: Comparison of MCP utilization with a comparably larger tool set (i.e., 100 tools) than Table 3. We report averaged performance over three test splits.

| Model | Tool | Param | AST |
|---|---|---|---|
| GPT-4o | 72.3 | 66.9 | 53.8 |
| Claude-4-Sonnet | 68.3 | 63.3 | 51.6 |
| Qwen3-4B | 61.7 | 59.8 | 49.0 |
| Llama3.1-8B | 39.8 | 37.7 | 23.5 |
| Groq-8B-Tool-Use | 2.9 | 1.4 | 1.3 |
| ToolACE-8B | 60.9 | 51.9 | 35.9 |
| MCP-Flow (Qwen-0.6B) | 64.7 | 63.4 | 51.6 |
| MCP-Flow (Qwen-4B) | **81.7** | **82.1** | **67.0** |

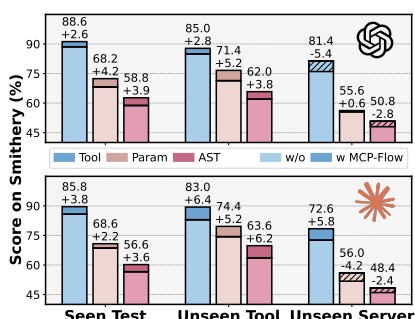

Figure 5: Comparing API model performance with and without retrieval augmented samples from MCP-Flow.

syntax tree (*AST*) (Patil et al., 2025), which strictly measures the generated function call format. In addition, we employ GPT-4o as the judge model to compute task success rate (*SR*) for experiments in Section 4.4. We also report efficiency metrics which quantify the average number of steps the agents incur to complete the requested tasks. Evaluation details are provided in Appendix C.2.

## 4.2 Training Small LLMs for MCP Tool Utilization

**Tool Selection and Formatting.** The first use of the MCP-Flow dataset is to train small models to master tool selection and formatting capabilities on real-world MCP servers. The task is, given a user query or instruction and a list of candidate MCP tools, for the model to directly generate a function call that both follows the MCP protocol and properly handles the user's request. We fine-tuned three backbone models of different sizes (i.e. Qwen3-0.6B, Qwen3-4B, and Llama3.1-8B) to accommodate different performance-efficiency trade-offs. Models are fine-tuned on all the training subsets.

**Undesired Performance of SOTA LLMs.** As shown in Table 3 and Table 4, current SOTA LLMs including Claude-4-Sonnet (Anthropic, 2024b) and GPT-4o (OpenAI, 2024) still have less than 60% AST accuracy on real-world MCP tools. When the candidate tool size are extremely large, i.e. 100, the results are worse, with Groq-8B-Tool-Use only achieve 3% accuracy.

Table 5: Evaluation results on GAIA. Weighted Step (WS) is computed using the token prices of the tested model and Qwen3-4B as weights. Using MCP-Flow to generate initial function call yields consistent improvements across models with varying capabilities and helps reduce overall costs.

| Backbone Model | Method | SR | SR Gain | Step | Weighted Step | WS Gain |
|---|---|---|---|---|---|---|
| Qwen3-4B | Base | 10.68 | – | 1.88 | 1.88 | – |
| | + MCP-Flow | 21.36 | +100% | 2.01 | 2.01 | –7% |
| GPT-4o | Base | 29.13 | – | 3.07 | 3.07 | – |
| | + MCP-Flow | 33.98 | +17% | 2.90 | 1.92 | +32% |
| Claude-4-Sonnet | Base | 55.34 | – | 6.01 | 6.01 | – |
| | + MCP-Flow | 57.28 | +4% | 6.28 | 5.29 | +12% |

**Superior Performance of MCP-Flow with Much Smaller Model Size.** In contrast, MCP-Flow achieves the best tool selection and formatting accuracy across different MCP marketplaces. Even tuned with a very small sized model, the output one can match or outperform much bigger competitor like Claude-4 and DeepSeek-V3. Other MCP datasets (MCPToolBench++) although with comparable data quality, are limited in scale and coverage and thus resulting very limited improvement.

**Out-of-Domain Generalization.** (1) Due to intrinsic differences among MCP servers, the *unseen-server* subset is harder, as nearly all models show performance drops when switching from *seen-test* to *unseen-server*. Conversely, the *unseen-tool* subset shares servers with *seen-test*, resulting in similar performance. (2) The fine-tuned MCP-Flow models demonstrate good generalization capabilities on *unseen-tool* and *unseen-server* and achieve marked improvement over the backbone models. (3) Tool-specialized models trained on traditional APIs still fall short in real-world MCP evaluation. In particular, Groq-8B-Tool-Use often predicts *"I can't help"*, which largely weakens its performance.

### 4.3 Enhancing Large LLMs on MCP Tool Utilization

For large closed-ended models that cannot be directly fine-tuned, our constructed dataset still proves highly useful through training-free distillation, serving as a retrieval database.

**Retrieval-Augmented Function Call.** Analogous to conventional Retrieval-Augmented Generation (RAG) paradigms, upon receiving a user instruction we first retrieve the top-$k$ (set to 5) most semantically similar data samples from our database and append them to the system prompt, thereby enhancing the model's ability to generate accurate function calls. The database of training instructions is constructed using embeddings produced by `Sentence-Transformers` (details in Appendix C.2). Retrieval is implemented via `Faiss-GPU`, employing cosine similarity as the ranking metric.

As illustrated in Figures 5 and 8 in Appendix: (1) Incorporating retrieved function-call exemplars consistently improves model performance, which underscores the continued value of our datasets. (2) We observe that Claude-4-Sonnet exhibits greater gains than GPT-4o, likely reflecting its stronger reasoning capacity; (3) Despite the inclusion of recalled samples, these large models still underperform relative to MCP-Flow, particularly on the *unseen-server* test set. This finding confirms the usefulness of fine-tuned, small-scale models for MCP function-call generation.

### 4.4 Enhancing LLM Agents on Agentic Tasks

**GAIA Benchmark.** GAIA (Mialon et al., 2023) is a challenging agentic benchmark that requires multi-step web searches and the proper use of external tools to successfully complete each test case. We demonstrate the third use case of MCP-Flow by generating the agent's initial function call and then allowing the agent to proceed based on this starting point. Details are discussed in Section C.3.

From Table 5 we conclude that: (1) **Enhanced tool-call tendency**: During our experiments, we observed that although GAIA tasks are highly challenging, models such as Qwen3 and GPT-4o often resist to invoking external tools and instead attempted to generate answers solely from their internal knowledge. However, such internal knowledge can be outdated or incomplete, thus introducing hallucination. By contrast, using our model to generate the initial function call helps mitigate this resistance to tool usage and guides the agent onto a path of more effective tool interactions. (2) **Reduced distraction from unavailable servers**: Leveraging our pre-constructed database allows us to filter out unavailable servers, thereby minimizing wasted attempts and reducing task failures. (3) These factors account for the observed improvements in performance and reductions in cost when replacing either expensive closed-ended models or weaker open-source models with MCP-Flow.

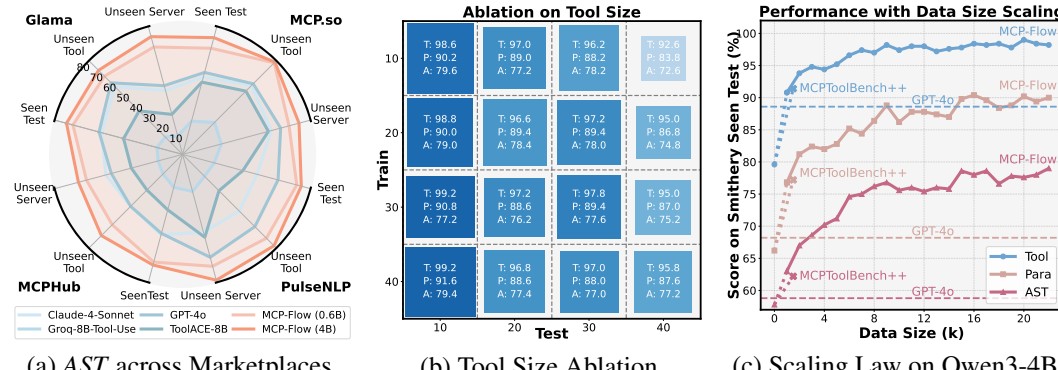

(a) *AST* across Marketplaces     (b) Tool Size Ablation     (c) Scaling Law on Qwen3-4B

Figure 6: Ablation results. (a) Model comparison across four platforms on three test splits. MCP-Flow significantly outperforms previous SOTA models. (b) The $(x, y)$ model is trained with tool size $y$ and tested with tool size $x$. Color darkness represents the *Tool* metric, while the width and height of the box represent *Param* and *AST*, respectively. (c) MCP-Flow features larger scale than the converted training version of MCPToolBench++ and provides better utility for tool-use training.

## 4.5 ABLATION STUDY

In this section, we conduct further ablation experiments to explore important aspects of tool use training. The experiment setup follows that of Section 4.2, with more results shown in Appendix B.3.

**Cross-Marketplace Comparison.** As shown in Figure 6.a: (1) Across all platforms, MCP-Flows consistently outperform all baseline models, which aligns with the results reported in Table 3. (2) The `MCPHub` and `Glama` datasets are more challenging than the `Smithery` dataset, likely due to the inclusion of less widely known servers. Such tools are more difficult for LLMs to predict because of their limited exposure to the Internet. (3) Although minor differences exist across marketplaces, these variations are insufficient to constitute severe data heterogeneity. This can be attributed to the fact that all marketplaces aggregate large-scale and diverse MCP servers, resulting in broadly comparable data distributions (also reflected by the dispersed embeddings shown in Figures 3.b and 7).

**Ablation on Candidate Tool Size.** We conduct experiments to investigate the effect of tool size (i.e., the number of candidate tools) on both model training and testing. Evaluations are performed on the *seen-test* split. As shown in Figure 6.b: (1) Model performance decreases on test sets with more candidate tools, which aligns with expectations since a larger number of tools increases task difficulty and may introduce distractions. (2) Models trained with larger tool sizes are more robust and perform better when evaluated with more candidate tools.

**Scaling Law Analysis.** Figure 6.c vividly illustrates that model performance increases with data size scaling. Among the three metrics, *Tool* accuracy quickly reaches a plateau, approaching nearly 100%, whereas the *AST* metric still shows potential for further improvement. Compared to MCPToolBench++, our constructed datasets not only offer better utility, as models improve more rapidly, but also encompass a much larger scale. Together, these two strengths make MCP-Flow currently the most effective dataset for training LLMs to master real-world MCP tools.

## 5 CONCLUSION

In this work, we introduce MCP-Flow, an automated pipeline, dataset and model suite designed to amplify LLM agents' utilization of real-world and rapidly evolving MCP servers and tools. MCP-Flow automates web-agent-driven server discovery from various MCP marketplaces and scalable data synthesis, producing a high-quality dataset with 60k+ samples covering 1,166 servers and 11,536 tools. We further develop compact fine-tuned models and a retrieval-augmented framework that significantly enhance LLMs in MCP tool selection, function call formatting, and multi-turn agentic tasks. Extensive experiments on standard benchmarks demonstrate that MCP-Flow consistently outperforms SOTA models and latest datasets, offering superior effectiveness at lower inference cost. Beyond model training and agentic enhancement, MCP-Flow lays the foundation for multi-dimension evaluation of MCP servers and tools, uncovering their heterogeneity and quality variations.

We believe MCP-Flow establishes a solid foundation for advancing real-world MCP tool capabilities for current LLM agents, and opens promising avenues for future research.

ETHICS STATEMENT

This research focuses on constructing a large-scale high-quality dataset to facilitate LLM agents in utilizing real-world diverse and continuously scaling MCP servers and tools.

The data are obtained from official marketplaces and websites, synthesizing using LLMs, or reprocessed versions of previously released datasets, with all sources and benchmarks properly cited. No discrimination, bias, or fairness issues are identified in this work. All collected information is stored locally in JSON format, which allows future research to operate without connecting to third-party servers and thereby avoids introducing external threats or security risks. Furthermore, our models only generate function calls and are not expected to produce potentially harmful content.

REPRODUCIBILITY STATEMENT

To ensure reproducibility, we provide all experimental setups and details in Section 4 and Appendix C. The dataset construction process and its statistics are described in Section 3 and Appendices D and E.1. Examples of the datasets are shown in Figure 4 as well as Tables 17 and 16. We have released initial samples in the anonymous repository and will make all data, source code, and model checkpoints publicly available upon acceptance of the paper.

THE USE OF LARGE LANGUAGE MODELS (LLMS)

In this paper, LLMs are primarily used for data synthesis and experiments in MCP-Flow. The process for data synthesis is described in Section 3, and the experimental procedures are detailed in Section 4, with additional information provided in the corresponding appendices. The prompts used are listed in Appendix E.1.

Apart from the usage in MCP-Flow, we also use LLMs to assist with paper writing. Specifically, we first draft the manuscript and then use LLMs (primarily OpenAI GPT models) to improve and revise the text. Afterward, all generated content is manually inspected, and we make final adjustments or rewrites as needed to ensure accuracy.

We additionally employ LLMs to support code writing for simple tasks, such as drawing figures and calculating statistics and use LLMs for repetitive tasks such as reformatting tables into LaTeX. All outputs are double-checked to ensure that no errors are introduced.

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

## A  CHALLENGES AND FUTURE DIRECTIONS

**Coverage of API-Required or Software-Specific MCP Servers.**  MCP servers that depend on specific software environments or require API keys represent two of the most significant challenges for automated MCP data construction. The deployment and configuration of certain software packages can be highly complex, demanding substantial manual effort even from experienced developers. In addition, the procedures for obtaining and applying API keys vary widely across server providers, making it difficult to standardize or fully automate this process.

Building on the automated data-construction pipeline of MCP-Flow, future work could extend coverage to these two types of servers, thereby enabling LLMs to master more diverse real-world MCP servers.

**Detection of Adversarial MCP Servers.**  The trustworthiness of MCP servers poses a critical challenge, as adversarial actors may maliciously upload or update servers on public marketplaces (Zhang et al., 2024; Song et al., 2025). Such adversarial servers can inject misleading information, exploit vulnerabilities in tool interfaces, or return deliberately manipulated outputs to mislead downstream agents. Detecting these malicious behaviors is inherently difficult because MCP servers often appear functionally similar to benign ones and may only exhibit malicious activity under specific triggers. Moreover, the lack of standardized auditing protocols and insufficient provenance tracking for server updates further exacerbate the risks of untrusted execution environments.

To the best of our knowledge, current MCP marketplaces do not implement explicit mechanisms to detect potentially malicious or harmful servers uploaded to their platforms. Future work may explore attack and defense strategies informed by the practices demonstrated in MCP-Flow.

**Evaluation of MCP Servers and Tools.**  During our investigation and experiments, we observe that current state-of-the-art agents (e.g., GPT, Claude) perform comparably in tool selection when the tools serve clearly different purposes. However, selecting between tools with similar functionalities remains challenging, as LLM agents rely primarily on tool descriptions, which may not accurately reflect the true quality or reliability of the tools.

Although we do not resolve this challenge in the present work, MCP-Flow provides a data platform that enables systematic investigation of this problem. As further discussed in Section B.1, our large-scale collection of MCP servers and systematic data construction establish a strong foundation for future research to explore this research direction.

Promising directions include (1) creating unified benchmarking datasets that test the same task across multiple tools with similar functionalities; (2) designing automated stress tests or scenario-based evaluations to capture stability, reliability, and latency under varying workloads; (3) introducing richer and more standardized metadata schemas, such as structured capability statements or performance profiles, to reduce ambiguity in tool descriptions; and (4) leveraging reinforcement learning (RL) or multi-agent comparison strategies to dynamically rank tools based on observed performance rather than static descriptions. Such efforts would enable more accurate tool selection, foster transparency, and ultimately improve the effectiveness of MCP ecosystems.

Table 6: Multi-dimension evaluation of MCP servers for weather tasks. Monthly tool calls are recorded from `Smithery` as of September 16, 2025. Although all servers target the same tasks, they differ in response quality, capability coverage, and efficiency. For instance, the United States Weather server balances response authenticity and token length but is limited to the U.S., while Weather API Server provides global coverage. Weather360 Server delivers comprehensive analyses, but its long responses may increase unnecessary cost and latency in multi-turn interactions.

| Server name | Performance | | Capability | | Efficiency | | Popularity |
|---|---|---|---|---|---|---|---|
| | SR | Quality | Feature | Coverage | Time(s) | Token | Monthly Call |
| Weather API Server | 76.9 | 3.27 | 3 | 5 | 2.40 | 35.2 | 2,400 |
| Weather Service | 46.2 | 2.82 | 1 | 5 | 2.66 | 11.0 | 1,802 |
| Weather360 Server | 76.9 | 3.92 | 5 | 5 | 2.62 | 7538.4 | 2,189 |
| Weather Server | 23.1 | 3.25 | 2 | 1 | 2.13 | 6596.2 | 2,849 |
| Weather Forecast Server | 84.6 | 3.45 | 5 | 5 | 2.21 | 225.6 | 5,292 |
| United States Weather | 38.5 | 4.25 | 6 | 1 | 1.93 | 372.0 | 67,400 |

## B   SUPPLEMENTARY EXPERIMENTS AND RESULTS

### B.1   EVALUATION OF MCP SERVERS

**Weather Domain Case Study.**   Beyond training LLMs for improved tool utilization, MCP-Flow also establishes the data foundation for the systematic evaluation of functionally similar MCP servers. To demonstrate this capability, we take the weather task as an case study and conduct a comparative analysis of six distinct weather-related MCP servers across multiple dimensions.

Note that with the large-scale crawling performed by MCP-Flow, our collected servers can support a wide range of task evaluations with numerous candidate servers and tools. For example, we have around 20 servers dedicated to weather tasks, providing a comprehensive playground for server-wise evaluation.

We use the instructions generated by MCP-Flow, as elaborated in Section 3.2, and randomly sample 13 test instructions from the pool across all tested servers to avoid bias toward any specific server. Subsequently, we deploy GPT-4o (OpenAI, 2024) as both the execution agent and evaluation judge. The agent attempts to resolve each constructed instruction using the tested MCP server's available tools, while the judge component assesses the quality and effectiveness of the responses. The prompt for judgments is provided in Section E.1.

**Multi-Dimension Evaluation.**   We evaluate each MCP server across four key dimensions, capturing its overall effectiveness and usability.

1. **Performance** covers the query success rate, abbreviated as *SR*, which measures the percentage of instructions receiving valid responses, and *Quality*, defined as the average score over successfully answered queries. The scoring mechanism employs a 0–5 point scale: instructions that fail to elicit any valid response receive zero points, while successful responses are scored from 1 to 5 based on comprehensiveness, accuracy, and practical utility. *Quality* thus represents the mean of all non-zero scores;

2. **Capability** assesses two aspects: *Feature* represents the number of available functions; and *Coverage* measures geographic applicability scored from one to five, where one indicates country-specific functionality and five indicates global coverage;

3. **Efficiency** measures both time consumption and token usage. *Time* refers to the average response latency in seconds, and *Token* denotes the average number of output tokens generated;

4. **Popularity** reflects real-world adoption, measured through the *Monthly Call* frequency on hosting platforms.

**Varying Characteristics and Performance across MCP Servers.**   As shown in Table 6, significant performance variations exist among functionally similar weather MCP servers. The Weather Forecast Server achieves the highest success rate (84.6%), while United States Weather demonstrates superior

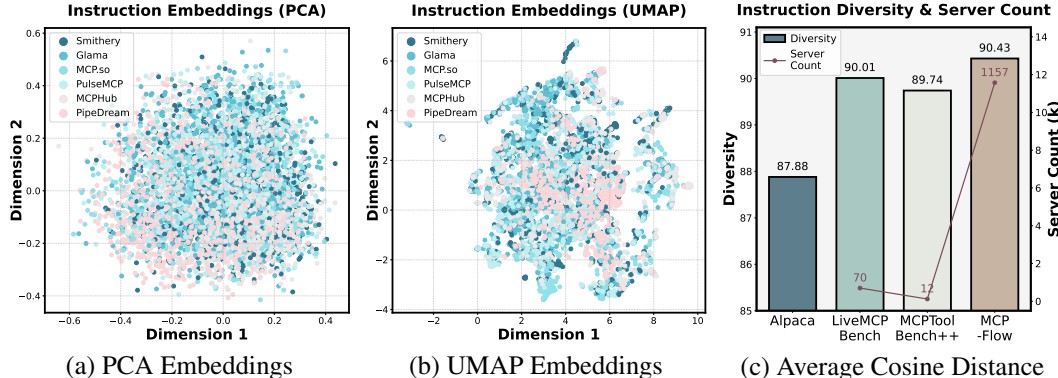

(a) PCA Embeddings      (b) UMAP Embeddings      (c) Average Cosine Distance

Figure 7: Visualization of instruction diversity. Across different dimensionality reduction techniques, MCP-Flow demonstrates a high diversity of instructions. Compared to other MCP datasets and benchmarks, MCP-Flow exhibits superior data scale and diversity, as measured by the MCP server count and average cosine distance.

response quality (4.25 average quality) and feature richness (6 points). Notably, *Feature* richness scores correlate strongly with average *Quality*, validating the hypothesis that MCP servers with more comprehensive and sophisticated tool ecosystems tend to deliver higher-quality responses

The United States Weather MCP, despite having the highest monthly usage (67,400 calls), shows moderate success rates (38.5%), indicating that popularity and technical performance may diverge. the disparity between popularity and technical performance metrics across different servers underscores the complex factors influencing user adoption, including ease of deployment, documentation quality, and ecosystem support beyond pure technical capabilities. These findings highlight the importance of systematic evaluation frameworks for MCP selection and ecosystem improvement.

## B.2 INSTRUCTION DIVERSITY

**Embedding Visualization.** We visualize instruction embeddings using several dimensionality reduction techniques, including PCA (Principal Component Analysis), t-SNE (t-distributed Stochastic Neighbor Embedding), and UMAP (Uniform Manifold Approximation and Projection), to comprehensively demonstrate the diversity of the MCP-Flow datasets. Specifically, PCA (Pearson, 1901) is a linear method that identifies the directions (principal components) capturing the maximum variance in the data, providing a straightforward global view of the embedding distribution. t-SNE (van der Maaten & Hinton, 2008), in contrast, is a nonlinear technique that excels at preserving local structure and revealing fine-grained clusters in high-dimensional data. UMAP (McInnes et al., 2018) combines the strengths of both linear and nonlinear methods, maintaining both local and global structures while being computationally efficient for large-scale embeddings.

The results in Figures B.2.a and B.2.b show that the embeddings are ubiquitous throughout the space, highlighting the diversity of instructions.

**Average Cosine Distance.** We also calculate diversity quantitatively by measuring pairwise cosine distances between instruction embeddings. We randomly sample 1,000 samples from MCP-Flow and each of the baselines. As shown in Figure 7.c, MCP-Flow achieves comparable diversity to human-written instructions from LiveMCPBench (Mo et al., 2025), demonstrating that our automated generation pipeline can produce instructions that match or even surpass human-crafted data in terms of variety.

## B.3 SUPPLEMENTARY ABLATION STUDY

**Cross-Marketplace Comparison.** We provide additional visualizations in the form of radar charts (Figure 9), illustrating other metrics for cross-marketplace performance comparisons. Detailed numerical results are presented in Section B.4.

Table 7: Comparison of MCP utilization capability on very large tool size (i.e. 100). We report averaged performance over three test splits.

| Model | Tool | Param | AST |
|---|---|---|---|
| GPT-4o-Mini | 69.4 | 66.5 | 52.8 |
| DeepSeek-V3 | 64.7 | 62.3 | 49.8 |
| Kimi-K2 | 65.7 | 62.7 | 49.6 |
| Qwen3-0.6B | 23.5 | 23.1 | 18.3 |
| MCPToolBench++ (0.6B) | 26.7 | 24.4 | 19.7 |
| MCPToolBench++ (4B) | 58.7 | 56.4 | 42.6 |
| MCP-Flow (Llama-8B) | 64.3 | 62.6 | 51.1 |
| MCP-Flow (Qwen-4B) | **81.7** | **82.1** | **67.0** |

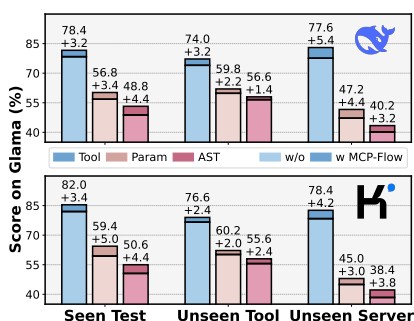

Figure 8: Comparing API model performance with and without retrieval enhanced samples from MCP-Flow.

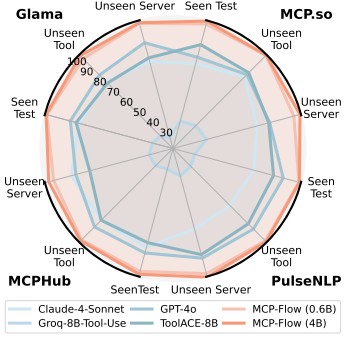

(a) *Tool* across Marketplaces

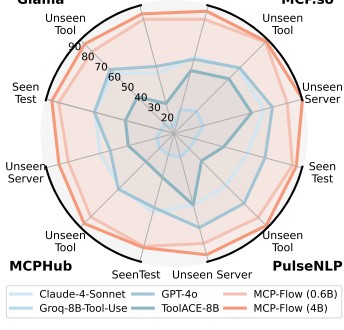

(b) *Param* across Marketplaces

Figure 9: Supplementary results comparing different models across various marketplaces. The *AST* results are shown in Figure 6, and the remaining two metrics are presented here. Note that the maximum values of the radial axes differ across figures. These results further prove the effectiveness of MCP-Flow compared to other baselines.

**Scaling Law Analysis.** For the scaling-law analysis, further results are shown in Figure 10. These charts demonstrate findings aligned with the conclusions drawn in Section 4.5. Compared with MCPToolBench++ (Fan et al., 2025), currently the only MCP effort that can be transformed into trainable samples, MCP-Flow offers both higher data quality and larger quantity. Notably, the MCPToolBench++ authors did not use these samples for model training, but focused solely on evaluation. This highlights the value of MCP-Flow as the only dataset capable of distilling practical knowledge about MCP servers into LLMs.

B.4 MCP TOOL SELECTION AND FORMAT EXPERIMENT ON VARIOUS MARKETPLACES

In this section, we present detailed experimental results covering nearly all marketplaces and models, with the complete results summarized in Table 8, Table 9, and Table 10. These findings reaffirm the primary conclusion discussed in Section 4.2: training small-scale models with MCP-Flow is effective to enhances their ability to utilize real-world MCP tools.

By comparing the performance of the same models across different platforms and test sets, we also identify additional observations and novel insights that extend our previous analysis: (1) Performance of MCPToolBench++: We notice that models trained on MCPToolBench++ perform relatively better when evaluated on MCP.so than other marketplaces. We attribute this to the fact that MCPToolBench++ is largely composed of servers from MCP.so, leading to a closer match in data characteristics. (2) Difficulty of the Glama *Unseen-Server* split: This split appears to be the most challenging, as most models perform poorly. For example, Groq-8B-Tool-Use achieves only 27.0% tool selection accuracy. (3) Challenges for API-based large models: They achieve particularly worse results on MCP.so than smaller models endowed with reasoning abilities. Our investigation reveals

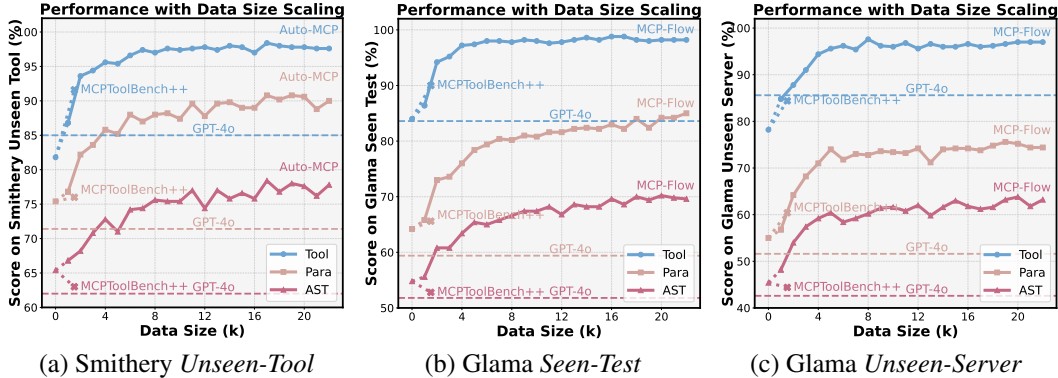

(a) Smithery *Unseen-Tool*      (b) Glama *Seen-Test*      (c) Glama *Unseen-Server*

Figure 10: Supplementary results for the scaling law analysis.

Table 8: Evaluation on subsets sourced from the `Glama` marketplace. Each model is evaluated on *Seen-Test*, *Unseen-Tool*, and *Unseen Server* test splits with *Tool* selection accuracy, *Param* format accuracy, and *AST* metrics.

| Category | Backbone Model | Seen Test | | | Unseen Tool | | | Unseen Server | | |
|---|---|---|---|---|---|---|---|---|---|---|
| | | Tool | Param | AST | Tool | Param | AST | Tool | Param | AST |
| **Large Models (> 10B) through Azure API** | | | | | | | | | | |
| Closed-Ended | GPT-4o | 83.6 | 59.4 | 51.8 | 82.2 | 64.2 | 60.6 | 85.6 | 51.6 | 42.6 |
| | GPT-4o-Mini | 83.6 | 60.6 | 50.4 | 78.6 | 61.4 | 57.2 | 81.6 | 46.2 | 39.2 |
| | GPT-4.1 | 71.8 | 52.8 | 44.2 | 66.4 | 56.0 | 50.6 | 72.6 | 47.8 | 39.0 |
| | Claude-4-Sonnet | 78.2 | 59.0 | 50.2 | 76.4 | 61.2 | 57.4 | 74.0 | 47.4 | 39.6 |
| Open-Ended | DeepSeek-V3 | 78.4 | 56.8 | 48.8 | 74.0 | 59.8 | 56.6 | 77.6 | 47.2 | 40.2 |
| | Kimi-K2 | 82.0 | 59.4 | 50.6 | 76.6 | 60.2 | 55.6 | 78.4 | 45.0 | 38.4 |
| **Small Models (≤ 10B) through Local Deployment** | | | | | | | | | | |
| General Model with Reasoning | Qwen3-0.6B | 59.4 | 40.4 | 33.0 | 55.4 | 34.4 | 26.6 | 64.2 | 39.0 | 33.2 |
| | Qwen3-4B | 84.0 | 64.2 | 54.8 | 78.4 | 67.4 | 56.2 | 78.2 | 55.0 | 45.4 |
| | Llama3.1-8B | 70.2 | 42.6 | 27.2 | 68.8 | 43.8 | 30.6 | 66.2 | 37.2 | 20.2 |
| Tool-Specialized | Groq-8B-Tool-Use | 32.6 | 18.0 | 15.8 | 30.6 | 18.2 | 14.8 | 27.0 | 13.4 | 11.4 |
| | ToolACE-8B | 80.0 | 40.2 | 36.6 | 75.4 | 41.0 | 36.8 | 76.2 | 28.6 | 24.8 |
| MCPToolBench++ | Qwen3-0.6B | 78.6 | 50.6 | 39.6 | 74.6 | 50.8 | 40.4 | 73.4 | 52.2 | 34.8 |
| | Qwen3-4B | 90.0 | 65.6 | 52.8 | 89.6 | 71.8 | 58.8 | 84.4 | 60.4 | 44.4 |
| MCP-Flow (Ours) | Qwen3-0.6B | 98.2 | 81.6 | 68.0 | 96.0 | 79.4 | 66.2 | 97.6 | 80.8 | 66.6 |
| | Qwen3-4B | 98.6 | 85.8 | 72.0 | 98.6 | 85.8 | 71.2 | 98.0 | 84.2 | 73.0 |
| | Llama3.1-8B | 98.6 | 84.6 | 71.2 | 97.8 | 85.6 | 73.2 | 98.0 | 83.0 | 72.0 |

that some servers from `MCP.so` lack formalized tool descriptions that strictly adhere to the OpenAI API protocol for tool invocation, which leads to frequent execution failures. This raises an important question about the robustness of LLM agents when interacting with less-formalized MCP tools.

## B.5 EFFICIENCY ANALYSIS

To evaluate the computational efficiency of the automated MCP server collection pipeline of MCP-Flow, we conduct a comprehensive analysis of resource and time consumption during the web agent-based crawling process. This analysis provides insights into the practical costs and scalability considerations of large-scale MCP server discovery.

Using `Smithery` as a representative example of all marketplaces, we track both token consumption and execution time for each MCP server collection attempt. Specifically, for every assistant turn during the collection process, we record input and output token usage of the GPT-4o model, including system prompts, user instructions, and prior tool interactions. We also record the wall-clock time

Table 9: Evaluation on subsets sourced from the `MCP.so` marketplace. Each model is evaluated on *Seen-Test*, *Unseen-Tool*, and *Unseen Server* with *Tool* selection accuracy, *Param* format accuracy, and *AST* metrics.

| Category | Backbone Model | Seen Test | | | Unseen Tool | | | Unseen Server | | |
|---|---|---|---|---|---|---|---|---|---|---|
| | | Tool | Param | AST | Tool | Param | AST | Tool | Param | AST |
| **Large Models (> 10B) through API** | | | | | | | | | | |
| Closed-Ended | GPT-4o | 76.6 | 56.0 | 50.8 | 82.6 | 65.6 | 59.8 | 80.0 | 71.0 | 60.8 |
| | GPT-4o-Mini | 75.4 | 55.0 | 46.6 | 81.8 | 66.4 | 58.4 | 85.0 | 73.4 | 60.6 |
| | Claude-4-Sonnet | 72.6 | 57.4 | 48.0 | 80.6 | 64.6 | 58.2 | 72.2 | 65.8 | 55.0 |
| Open-Ended | DeepSeek-V3 | 77.0 | 59.8 | 51.6 | 83.2 | 65.0 | 60.8 | 83.4 | 72.2 | 62.2 |
| | Kimi-K2 | 77.0 | 59.6 | 50.8 | 81.0 | 64.4 | 57.2 | 74.6 | 66.2 | 56.4 |
| **Small Models (≤ 10B) through Local Deployment** | | | | | | | | | | |
| General Model with Reasoning | Qwen3-0.6B | 59.8 | 42.2 | 35.4 | 59.0 | 42.4 | 39.0 | 63.0 | 44.8 | 36.6 |
| | Qwen3-4B | 85.8 | 68.4 | 57.2 | 87.2 | 72.0 | 62.2 | 81.8 | 72.0 | 59.6 |
| | Llama3.1-8B | 74.4 | 69.5 | 44.1 | 73.2 | 69.6 | 45.8 | 71.0 | 63.3 | 38.6 |
| Tool-Specialized | Groq-8B-Tool-Use | 37.0 | 24.8 | 20.6 | 39.4 | 28.8 | 27.0 | 40.8 | 28.2 | 24.0 |
| | ToolACE-8B | 84.4 | 49.0 | 44.8 | 84.2 | 56.4 | 53.6 | 79.4 | 58.0 | 53.2 |
| MCPToolBench++ | Qwen3-0.6B | 78.0 | 53.4 | 42.8 | 85.8 | 64.6 | 52.6 | 79.8 | 63.8 | 48.4 |
| | Qwen3-4B | 91.6 | 68.8 | 54.2 | 94.8 | 78.2 | 66.0 | 91.2 | 78.8 | 62.6 |
| MCP-Flow (Ours) | Qwen3-0.6B | 97.2 | 80.8 | 65.8 | 99.0 | 87.4 | 77.6 | 95.0 | 80.2 | 66.0 |
| | Qwen3-4B | 99.0 | 85.8 | 72.4 | 99.0 | 88.0 | 78.2 | 98.8 | 89.6 | 72.2 |
| | Llama3.1-8B | 99.2 | 84.8 | 72.0 | 99.4 | 88.2 | 78.0 | 97.2 | 87.6 | 72.2 |

Table 10: Evaluation on subsets sourced from the `MCPHub` marketplace. Each model is evaluated on *Seen-Test*, *Unseen-Tool*, and *Unseen Server* test splits with *Tool* selection accuracy, *Param* format accuracy, and *AST* metrics.

| Category | Backbone Model | Seen Test | | | Unseen Tool | | | Unseen Server | | |
|---|---|---|---|---|---|---|---|---|---|---|
| | | Tool | Param | AST | Tool | Param | AST | Tool | Param | AST |
| **Large Models (> 10B) through API** | | | | | | | | | | |
| Closed-Ended | GPT-4o | 84.8 | 56.0 | 49.0 | 86.4 | 57.2 | 44.0 | 80.2 | 49.6 | 44.2 |
| | GPT-4o-Mini | 81.0 | 51.4 | 44.2 | 83.2 | 59.6 | 44.0 | 76.4 | 52.6 | 41.8 |
| | Claude-4-Sonnet | 81.4 | 58.8 | 49.8 | 80.6 | 56.0 | 44.0 | 83.0 | 56.0 | 47.6 |
| Open-Ended | DeepSeek-V3 | 80.4 | 53.2 | 46.8 | 87.4 | 58.0 | 44.0 | 74.8 | 52.2 | 41.0 |
| | Kimi-K2 | 78.4 | 54.2 | 46.6 | 83.4 | 56.4 | 43.2 | 70.3 | 47.7 | 40.6 |
| **Small Models (≤ 10B) through Local Deployment** | | | | | | | | | | |
| General Model with Reasoning | Qwen3-0.6B | 71.4 | 43.4 | 24.8 | 74.8 | 51.6 | 20.6 | 71.2 | 46.8 | 28.8 |
| | Qwen3-4B | 81.8 | 59.2 | 49.6 | 86.2 | 62.6 | 47.6 | 76.2 | 56.0 | 42.4 |
| | Llama3.1-8B | 70.4 | 43.4 | 24.8 | 73.4 | 50.4 | 21.4 | 68.6 | 44.0 | 27.8 |
| Tool-Specialized | Groq-8B-Tool-Use | 32.8 | 23.2 | 20.4 | 34.6 | 22.2 | 16.6 | 34.8 | 21.0 | 15.2 |
| | ToolACE-8B | 78.4 | 37.2 | 34.4 | 80.8 | 33.8 | 30.2 | 71.6 | 38.4 | 31.8 |
| MCPToolBench++ | Qwen3-0.6B | 84.4 | 57.2 | 41.0 | 83.4 | 55.2 | 33.2 | 79.2 | 53.6 | 34.6 |
| | Qwen3-4B | 92.2 | 62.8 | 46.0 | 93.6 | 65.0 | 38.0 | 90.6 | 68.4 | 44.8 |
| MCP-Flow (Ours) | Qwen3-0.6B | 96.2 | 80.0 | 66.8 | 97.4 | 80.6 | 59.6 | 94.0 | 75.6 | 59.6 |
| | Qwen3-4B | 98.0 | 80.8 | 68.6 | 98.2 | 86.2 | 68.8 | 97.0 | 81.4 | 64.6 |
| | Llama3.1-8B | 98.0 | 83.2 | 71.6 | 97.8 | 87.2 | 68.2 | 97.4 | 82.0 | 65.6 |

from the initiation to the completion of each MCP server's detailed page scraping. For each MCP collection attempt, input and output tokens were aggregated across all turns, and the final metrics represent averages computed over all MCP attempts that successfully produced MCP configurations.

Table 11: Efficiency metrics for automated MCP server collection from `Smithery`. Averages are computed over successful collection attempts that produce MCP server configurations.

| Efficiency Metric | Average Value per Server |
|---|---|
| Input Tokens | 97,827 |
| Output Tokens | 414.39 |
| Execution Time (s) | 42.17 |
| Price (US cent ¢) | 24.87 |

Table 12: Key training parameters regarding optimization, data and efficiency.

| Parameter | Value | Parameter | Value |
|---|---|---|---|
| **Optimization** | | | |
| Device Number | 2 | Batch Size | 2 |
| Gradient Accumulation Steps | 8 | Learning Rate | 5e-5 |
| LR Scheduler Type | cosine | Warmup Ratio | 0.1 |
| **Data** | | | |
| Preprocessing Workers | 16 | Dataloader Workers | 4 |
| D-type | BF16 | Max Length (Input+Output) | 8192 |
| **Efficiency** | | | |
| LoRA Rank | 16 | LoRA Alpha | 32 |
| DeepSpeed Stage | 0 | Load in 8bit | False |

Experimental results summarized in Table 11 demonstrate that the collection of newly updated servers is cost-effective. For instance, crawling 100 latest servers incurs a cost of approximately 2 dollars. We reiterate that, based on our large-scale collection, future work only needs to carry out incremental crawling, instead of re-collecting the entire marketplaces.

## C   EXPERIMENT DETAILS

In this section, we provide detailed information about the our experiments, including parameters, setups and metrics. Training and inference details are presented in Section C.1. Evaluation details are presented in Section C.2. Details related to the GAIA benchmark are presented Section C.3.

### C.1   TRAINING AND INFERENCE DETAILS

**Training Framework and Parameters.**   We adopt LLaMA-Factory (Zheng et al., 2024) to fine-tune our local models, as it is widely used and supports training of the latest model series, including Qwen3 and Llama3. For smaller models, we employ LoRA fine-tuning (Hu et al., 2022) due to its superior resource efficiency.

To further reduce GPU memory consumption, we utilize DeepSpeed (Ren et al., 2021), setting the Zero Redundancy Optimizer (ZeRO) stage to 0, following LLaMA-Factory's implementation. Other key training parameters are summarized in Table 12.

**Inference Parameters.**   We employ vLLM (Kwon et al., 2023) to accelerate inference and deploy the local model on a single H100 GPU. Other parameters are presented in Table 13.

In Section 4.2, for Qwen base models, we adopt the non-thinking template to ensure a fair comparison, as all models are evaluated without additional reasoning techniques. In Section 4.2, all models are evaluated using the thinking template, since the evaluation is conducted on a challenging agentic benchmark.

Table 13: Key inference parameters regarding data generation and vLLM.

| Parameter | Value | Parameter | Value |
|---|---|---|---|
| **Generation** | | | |
| Do Sample | True | Temperature | 0.7 |
| Top P | 0.8 | Max Tokens (Output) | 4096 |
| **vLLM** | | | |
| Max Length (Input+Output) | 32768 | Enforce Eager | True |

Table 14: Models used in the experiments. Details about the API model versions and the specific Hugging Face URLs for the locally deployed models are presented.

| Azure API | | Local Deployment | |
|---|---|---|---|
| Model Name | Version | Model Name | Hugging Face URL |
| GPT-4o | gpt-4o-2024-11-20 | Qwen3-0.6B | Qwen/Qwen3-0.6B |
| GPT-4o-Mini | gpt-4o-mini-2024-07-18 | Qwen3-4B | Qwen/Qwen3-4B |
| Gemini-2.5-Pro | gemini-2.5-pro | Qwen3-8B | Qwen/Qwen3-8B |
| Claude-4-Sonnet | claude-4-sonnet | Llama3.1-8B | meta-llama/Llama-3.1-8B-Instruct |
| DeepSeek-V3 | deepseek-v3-0324 | Groq-8B-Tool-Use | Groq/Llama-3-Groq-8B-Tool-Use |
| Kimi-K2 | kimi-k2-0905-preview | ToolACE-8B | Team-ACE/ToolACE-8B |

**Model Version.** As shown in Table 14, for reproducibility and fair comparison, we provide a detailed description of our models. All API-based models are accessed through Microsoft's Azure platform[1], while the relatively small open-source models are downloaded from Hugging Face.

**Environment and Resources.** For the MCP client and server deployment, the tool responses were obtained in a macOS environment with Node.js. We configured the path to the current workspace directory. All training and evaluation experiments were conducted on a Linux server equipped with eight NVIDIA H100-SXM-80GB GPUs.

Training the Qwen3-4B models on the full function call dataset for two epochs take approximately 12 hours on 2 GPUs.

## C.2 EVALUATION DETAILS

**Metrics for Tool Selection and Formatting.** For tool selection and format evaluation, we adopt three metrics with increasing strictness following representative prior work (Wu et al., 2024; Shen et al., 2024; Liu et al., 2024a; Patil et al., 2025; Gao et al., 2025).

1. **Tool selection accuracy** (*Tool*): This metric measures the correctness of tool selection by calculating the percentage of predicted tool names that match the ground-truth tool names.

2. **Parameter format accuracy** (*Param*): This metric evaluates the model's ability to generate correctly formatted tool parameters. Each predicted parameter name is compared recursively with the corresponding ground-truth parameter, without requiring positional alignment. The evaluation follows an all-or-nothing rule: if any ground-truth parameter is unmatched, the entire prediction is considered incorrect. This ensures that the model identifies all required parameters, regardless of order.

3. **Abstract Syntax Tree** (*AST*): AST is adopted from BFCL (Patil et al., 2025). According to the authors, this metric exhibits a strong alignment with actual execution results. A function call is deemed correct if the function name matches exactly and all parameter values fall within their respective allowed sets. For further details on the AST matching rules, please refer to Patil et al. (2025).

---

[1] https://azure.microsoft.com/en-us/pricing/details/cognitive-services/openai-service/

**Metrics for Evaluating MCP Servers.**

1. **Query Success Rate:** Measures the percentage of queries that receive non-zero scores, serving as an indicator of the MCP's functional robustness and universal applicability. Higher success rates suggest that the MCP can handle a broader range of weather-related queries effectively.

2. **Average Performance:** Calculates the mean score across all successful queries, reflecting the professional quality and effectiveness of the MCP's responses when it functions correctly.

3. **Feature Richness:** Evaluates the comprehensiveness and sophistication of each MCP's tool ecosystem. For this assessment, we employ a comparative scoring methodology where each individual tool within an MCP is evaluated against functionally similar tools across all weather MCPs in our dataset. Each tool receives a score from 1-5 based on two primary criteria: (1) the level of detail and granularity in its functionality, and (2) the breadth of its applicability and use case coverage. Tools offering basic weather information retrieval receive lower scores, while those providing advanced features such as multi-location forecasting, historical data analysis, or specialized meteorological computations receive higher scores. The final Feature Richness score for each MCP represents the sum of scores across all its constituent tools, providing a quantitative measure of the server's overall functional depth and versatility.

4. **Efficiency Metrics:** We measure Average Execution Time to assess the computational responsiveness of each MCP, while Average Output Token metrics quantify the communication overhead and resource consumption associated with each interaction. These efficiency metrics are particularly crucial for production deployments where latency and cost considerations significantly impact user experience and system scalability.

5. **Monthly Tool Calls:** Captures real-world adoption patterns by measuring the frequency of user interactions with each MCP on its respective hosting platform. This metric serves as a proxy for community acceptance and practical utility, as user preference patterns often reflect the perceived value and reliability of different MCP implementations.

**Instruction Diversity and Embedding Details.** We employ the python package `Sentence-Transformers`[2] to compute instruction embeddings. This setup is applied for similarity filtration in Section 3.3, retrieval augmentation in Section 4.3, and instruction diversity analysis in Section B.2. For filtering and retrieval, we adopt the widely used model `mxbai-embed-large-v1`[3], while `all-MiniLM-L6-v2`[4] is employed for diversity computation.

To ensure reproducibility, we provide a concise code snippet illustrating how embedding similarity is calculated:

```
from sklearn.metrics.pairwise import cosine_similarity
model = SentenceTransformer(model_path, device="cuda", trust_remote_code=True)
embeddings_1 = model.encode(sentence1, max_length=512, task="text-matching")
embeddings_2 = model.encode(sentence2, max_length=512, task="text-matching")
similarity = cosine_similarity([embeddings_1], [embeddings_2])
```

## C.3 GAIA BENCHMARK DETAILS

**Setups.** For the GAIA benchmark evaluation, we carefully select several powerful web-search MCP tools, including Google Search using Serper API, Jina Web Parser, and Firecrawl.

Following common practice in agentic research (Wu et al., 2025; Li et al., 2025), we adopt the Pass@1 metric for evaluation. All experiments are conducted on the text-only validation subset[5], which comprises 103 questions. To ensure a fair comparison and minimize potential confounding effects arising from tool heterogeneity, we disable Firecrawl during the experiments. We also cap the maximum number of execution steps at 10, a limit that is rarely approached in practice. Furthermore, to prevent excessive context accumulation across multiple turns in web parsing, we employ GPT-4.1-nano to summarize the retrieved content before subsequent processing. To prevent the agent

---

[2]https://github.com/UKPLab/sentence-transformers

[3]https://huggingface.co/mixedbread-ai/mxbai-embed-large-v1

[4]https://huggingface.co/sentence-transformers/all-MiniLM-L6-v2

[5]https://github.com/sunnynexus/WebThinker/blob/main/data/GAIA/dev.json

from exploiting trivial shortcuts, such as directly querying answers related to GAIA, we employ a keyword-filtering mechanism that removes terms including "GAIA" and "HuggingFace" from the search inputs, while still allowing general external resource access.

**Metrics.** For the evaluation of models on the GAIA benchmark, we adopt the official success rate metric to assess performance, and use two additional metrics, step number and weighted step number, to evaluate efficiency.

1. **Success Rate** (*SR*): *SR* is computed using an LLM-as-a-judge approach, comparing the agent's final answer with the ground-truth label. The evaluation prompt is provided in **??**. Specifically, we use GPT-4o as the judge model. For the third label, *partially correct*, where the judge model is uncertain about correctness, we manually verify whether the ground-truth label is included in the answer. For example, an answer of "INT. THE CASTLE - DAY" is considered correct with the ground truth "THE CASTLE", even though GPT-4o notes: *"The model's answer includes additional detail ('INT.' and '- DAY') that is not part of the ground truth answer ('THE CASTLE'). While the core location is correct, the format does not exactly match the ground truth."*

2. **Step Number**: This metric directly computes the average number of assistant messages in a trajectory. It accounts for function calls without semantic content, direct textual responses, intermediate reasoning steps, and the final answer.

3. **Weighted Step Number** (*WS*): Since our tuned function-call model is considerably smaller than typical LLM agents, employing MCP-Flow to initiate function calls substantially reduces cost. We use the API input-token price difference as the weighting factor to compute a weighted step number. The model price of MCP-Flow is based on the official pricing of Qwen3-4B[6], and we assume an exchange rate of 7 Chinese yuan to 1 US dollar for estimation.

# D DATASET CONSTRUCTION DETAILS

In this section, we detail the implementation of the automated data construction pipeline of MCP-Flow, drawing on the server collection described in Section D.1 and the marketplace introduction provided in Section D.2.

## D.1 AUTOMATED SERVER AND TOOL COLLECTION DETAILS

**Web Agent.** For Smithery, MCPHub, Glama, MCP.so, and PipeDream, the server collection process follows a largely unified procedure. We employ the web agent based on Playwright MCP[7] to (1) navigate to the home page, (2) collect all listed server information (including names, descriptions, and IDs), (3) click on each server entry, (4) use the tool snapshot function to capture information and extract the MCP server configuration from the current page in JSON format, and (5) proceed to the next page and repeat steps (2)–(4).

Glama differs in that it does not provide a single homepage listing all servers. Instead, its homepage allows repeatedly clicking "Load More," which yields only about 30 servers, a quantity insufficient for our purposes. We therefore adapted our approach by navigating to multiple search-result pages using query keywords such as `https://glama.ai/mcp/servers?query=a&sort=github-stargazers%3Adesc`, and collected all servers listed on these pages. To prioritize the most popular servers, we sorted the results by GitHub star counts.

For MCPHub, to maximize efficiency we selected the homepage `https://mcphub.com/online-hosted-servers`, which lists all servers hosted through MCPHub endpoints. These servers typically provide explicit server configurations, unlike many others on the platform.

For MCP.so, to improve efficiency and avoid redundant crawling, we directly leveraged pre-crawled server information from MCPCorpus (Lin et al., 2025), supplementing it with additional tool information to generate function call.

---

[6]`https://help.aliyun.com/zh/model-studio/models?spm=a2ty02.30268951.d_model-market.17.71dc74a1GUEilx#2c9c4628c9yyd`

[7]`https://github.com/microsoft/playwright-mcp`

---

**Algorithm 1** Automated MCP Server Collection from Marketplaces

---

**Require:** Marketplace URLs: $\mathcal{M} = \{M_1, M_2, ..., M_k\}$
**Ensure:** Deduplicated server configurations $\mathcal{S}$ and tool information $\mathcal{T}$
 1: Initialize empty sets: $\mathcal{S}_{raw} \leftarrow \emptyset, \mathcal{S} \leftarrow \emptyset, \mathcal{T} \leftarrow \emptyset$
 2: **for** each marketplace $M_i \in \mathcal{M}$ **do**
 3:     Initialize Playwright web agent
 4:     Navigate to marketplace homepage $M_i$
 5:     **while** more pages available **do**
 6:         Extract server list from current page using MCP List Collection Prompt
 7:         **for** each server $s$ in server list **do**
 8:             Navigate to server detail page
 9:             Extract JSON configuration using Server Configuration Extraction Prompt
10:             $\mathcal{S}_{raw} \leftarrow \mathcal{S}_{raw} \cup \{s\}$
11:         **end for**
12:         Navigate to next page
13:     **end while**
14: **end for**
15: **// Server Deduplication**
16: **for** each server $s_i \in \mathcal{S}_{raw}$ **do**
17:     Extract tool descriptions $D_i$ from $s_i$
18:     **if** $\nexists s_j \in \mathcal{S}$ such that $D_i = D_j$ **then**
19:         $\mathcal{S} \leftarrow \mathcal{S} \cup \{s_i\}$
20:     **end if**
21: **end for**
22: **// Local Deployment and Tool Collection**
23: **for** each server $s \in \mathcal{S}$ **do**
24:     Deploy server locally using MCP client (npm/uvx for stdio, URL for SSE)
25:     Extract tool information: name, description, input schema
26:     $\mathcal{T} \leftarrow \mathcal{T} \cup \{\text{tools from } s\}$ deployment failure
27:     Mark server as unavailable and continue
28: **end for**
29: **return** $\mathcal{S}, \mathcal{T}$

---

**Python SDK.** MCP-Marketplace[8] implements a python SDK which provides direct post-get approach for obtaining server information. Our code snippet is provided below:

```python
import mcp_marketplace as mcpm

# Select data source
mcpm.set_endpoint("deepnlp")

# Query MCP Marketplace
result_q = mcpm.search(mode="list", page_id=0, count_per_page=100)

# Extract information
for item in result_q["items"]:
    item_id = item["id"]
    item_name = item["content_name"]
    item_description = item["description"]
    item_url = item["detail_url"]
```

We note that the agent method is also applicable to these two endpoints. We employ Python SDK as a cheaper method as we crawled 10,000 servers and conduct filtration.

For `PipeDream` the difficulty lies in that almost all servers from this endpoint requires personalized API key for deployment which, as we have previous elaborate, is of great difficulty for automated collection. However, after some digging and cross-comparison, we find that PipeDream demon-

---

[8]https://pypi.org/project/mcp-marketplace/

Table 15: List of all marketplaces utilized in this paper. In the main paper, we treat `PulseMCP` and `DeepNLP` as the same market source, as the Python SDK collection method supports both. *Server Host* indicates whether the marketplace self-hosts any MCP servers and provides proxy access to them. Server counts are recorded as of 2025.09.16.

| Icon | Marketplace | Home Page | Server Host | Server Count |
|---|---|---|---|---|
| | Smithery | smithery.ai | ✓ | 6859 |
| | Glama | glama.ai/mcp/servers | ✗ | 9361 |
| | MCP.so | mcp.so | ✗ | 16563 |
| | MCPHub | mcphub.com | ✓ | 27793 |
| | PulseMCP | pulsemcp.com/servers | ✗ | 6073 |
| | DeepNLP | deepnlp.org/store/mcp-server | ✗ | 11k+ |
| | PipeDream | mcp.pipedream.com | ✓ | 2877 |

state tool information for almost all servers on corresponding page and each server is linked to its GitHub directory on the PipeDream official repository. For example, `https://github.com/PipedreamHQ/pipedream/tree/master/components/notion` contains information for the Notion Server. Each tool has a seperate directory under "/actions". And each tool has a js file which specifies its name, description and parameters. Based on this discovery, we implement a crawling agent to collect servers and tool information from the whole website and obatin around 1,500 servers and filter the 100 most popular.

## D.2 MARKETPLACE INTRODUCTION

We provide a brief introduction to each of the six marketplaces we utilize.

1. **Smithery**[9] is an emerging platform that standardizes the integration of external services into large language models and autonomous agents via the Model Context Protocol (MCP). It lowers deployment and maintenance costs by providing a centralized registry, development tool chains, and hosting infrastructure, thereby promoting reusability and interoperability. However, its adoption also requires careful attention to security, privacy, and version control.

2. **Glama** is a platform that provides discovery, indexing, and connectivity for MCP servers, clients, and tools. It enables users to search, compare, and access thousands of MCP servers through multiple transports, as well as via an API gateway or chat-UI. Servers are ranked along dimensions such as security, compatibility, and usability, helping users choose the right ones.

3. **MCP.so** is a community-driven platform that collects and organizes third-party MCP Servers. It serves as a central directory where users can discover, share, and learn about various MCP Servers available for AI applications.

4. **MCPHub** is a central platform for discovering, testing, and integrating Model Context Protocol (MCP) servers. It allows AI assistants to securely connect with external data sources and tools, extending their capabilities beyond their training data. Users can browse detailed server documentation, test servers in an online inspector, and seamlessly integrate them into their applications.

5. **PipeDream** offers a dedicated MCP server that integrates thousands of applications and pre-built tools through a standardized interface. It allows large language models and AI assistants to securely invoke external APIs and perform real-world tasks using managed OAuth and encrypted credential storage. This setup streamlines authentication and interaction patterns, enabling scalable, secure, and protocol-compliant access to a wide range of services.

[9]`https://smithery.ai/`

Table 16: A list of example MCP Servers. We present details about the server names, corresponding URLs, source marketplaces, and descriptions.

| Name | Marketplace | Category | Description |
| --- | --- | --- | --- |
| Youtube Transcript | Smithery | Art & Entertainment | Retrieve transcripts of YouTube videos. |
| Gen-PDF Server | Smithery | File System | Generate professional PDF documents from markdown content with customizable styling options including dark mode and advanced page settings. Convert any github flavored markdown to high-quality PDFs using the Gen-PDF API. |
| YGO Chinese Card Database | Smithery | Database | Provide fast and easy access to Chinese Yu-Gi-Oh! card information and images through keyword search and ID queries. Integrate seamlessly with your applications to retrieve detailed card data and visuals. Support both stdio and Streamable HTTP modes for flexible deployment. |
| ESA MCP Server | Glama | Web Search & Data | API through the Model Context Protocol, supporting article search and retrieval with a compliant MCP interface. |
| Deepwiki MCP Server | Glama | Developer Tools | An MCP server that fetches and converts Deepwiki documentation into Markdown, allowing users to crawl pages from deepwiki. |
| code-runner-mcp | MCP.so | Developer Tools | Run JavaScript/Python code in a secure sandbox with support for **any package import**." |
| Get My Location | MCP.so | Map & Weather | Get My Location is a location acquisition server that retrieves the precise current location of the user through browser authorization, integrating with weather and map services for enhanced functionality. |
| Crypto Price & Market Analysis MCP Server | MCP.so | Finance | A Model Context Protocol (MCP) server that provides comprehensive cryptocurrency analysis using the Coin-Cap API. This server offers real-time price data, market analysis, and historical trends through an easy-to-use interface. |
| browser-scraper | MCPHub | Browser Automation | The browser-scraper MCP is designed to facilitate web scraping using a browser, providing content in Markdown format. |
| google-news-search | MCPHub | Web Search & Data | Aigeon AI Google News Search is a Python-based server application designed to interact with the Google News search engine via the SerpApi. |
| AutoBlogger | PipeDream | Art & Entertainment | Automatically create and publish posts Set it up once and then redirect your focus to more important tasks. |
| Todoist | PipeDream | File System | Todoist is a delightfully simple yet powerful task planner and to-do list app. |
| travel-planner | Manual | Map & Weather | A Travel Planner Model Context Protocol (MCP) server implementation for interacting with Google Maps and travel planning services. This server enables LLMs to perform travel-related tasks such as location search, place details lookup, and travel time calculations. |

## D.3 DATA GENERATION DETAILS

**Slot-Fill Revision.** As part of the regular-expression rules outlined in Section 3.2, we detect three types of placeholders: file names, URLs, and directories. For file names, we substitute the placeholders with valid local files based on their extensions. For URLs, we replace them with

Table 17: Example test instructions used to evaluate the quality and functionality of weather-related MCP servers, as discussed in Section B.1.

| Instruction | Test Purpose |
| --- | --- |
| What's the current temperature at 40.7128,-74.0060? | Temperature query with coordinates |
| Are there any active weather alerts near 29.7604,-95.3698? | Weather alerts query with coordinates |
| Show current weather alerts for TX. | Weather alerts query by state abbreviation |
| Provide current weather conditions at -12.0464,-77.0428. | General weather conditions with coordinates |
| What is the current weather in Gweru, Zimbabwe? | Weather query by city and country name |

real-world examples collected from GitHub[10] . For directories, we normalize them to the absolute path of our local working directory.

To preserve anonymity and protect the authors' privacy, we substitute all local file names and directories with specific placeholders and provide simple code that allows other researchers to replace them with customized input.

# E  PROMPTS AND EXAMPLES

## E.1  PROMPT TEMPLATES

---

**Prompt 1: MCP Server List Collection Prompt**

Use the tool Playwright to obtain the information about listed mcp servers in a json format from the given url.
A mcp server is a tool that can be used to interact with the system, such as "Exa Search", "xxx MCP Server".
The mcp server name is usually contained in the "heading".
The description of the mcp server is usually contained in the "paragraph" near the "heading".

Url: {url}&page={page}

Only need to return the json data, no other text.
Only need the names and descriptions of the mcp servers.
Prefer browser_snapshot than browser_evaluate.

---

Figure 11: Prompt for automated MCP server discovery across marketplace pages. This prompt instructs the web agent to systematically collect server names and descriptions from marketplace listings, supporting the large-scale server collection described in Section 3.1.

---

[10]https://gist.github.com/bejaneps/ba8d8eed85b0c289a05c750b3d825f61#file-websites-csv

---

**Prompt 2: Smithery Server Configuration Extraction Prompt**

Use the tool Playwright to obtain the json data of the given mcp server.
You need to follow the steps below:
1. Open the browser: {url}&page={page}
2. Click the corresponding mcp server {mcp_name}.
3. Click button "JSON" at the right side of the new page.
4. Click the "Connect" button poped up.
5. Retrieve the json data from the current page which specify how to install the mcp server.

Only need to return the json data that contains "mcpServers" and "command".
Prefer browser_snapshot than browser_evaluate.
Don't click on "Generate URL" button!

---

Figure 12: Prompt for extracting MCP server configuration files from Smithery marketplace. This prompt guides the web agent through the specific navigation steps required to access and retrieve JSON configuration data for individual servers, as part of the automated server collection pipeline.

---

**Prompt 3: Tool-based Instruction Generation Prompt**

You are given a specific tool from a mcp server. You need to generate an instruction which requires to utilize this tool.
Each instruction needs to use exactly {number} tools belong to the mcp server.

## Input
- **MCP Server information**:
[MCP Server Name] {mcp_name}
[MCP Server Description] {mcp_description}
- **Tool information**:
[Tool Name] {tool_name}
[Tool Description] {tool_description}
[Tool Schema] {tool_schema}

## Requirement
The instruction should not directly include the name of the mcp server or the name of the tools.
The instruction must not look similar to the tool description.
Make sure The tool and instruction in your output are aligned.
Try to improvise and return 5 instruction candidates.

## Example
{example}

## Output Format
[Instruction1] <your generated instruction>
[Instruction2] <your generated instruction>
[Instruction3] <your generated instruction>
[Instruction4] <your generated instruction>
[Instruction5] <your generated instruction>

---

Figure 13: Prompt for tool-based few-shot instruction generation. This prompt ensures instructions are naturally formulated without directly mentioning tool names, as described in the tool-based few-shot generation stage of Section 3.2.

**Prompt 4: Slot-Fill Revision Prompt**

You are an expert at identifying missing but necessary details.
You will be provided with a tool's parameters and a user query. Your task is to supplement any missing information required by the tool in the user query.

If the query lacks required details, retrieve them from the provided environmental context. Then, revise the user query by adding the appropriate missing details.
Ensure that your revisions are realistic and reasonable.

## Requirements
1. Be realistic and authentic, stick to the given environmental context if given.
2. For not included details in the environmental context, like place, date and institutions, etc, try to use real-world names; if they don't affect the common knowledge, you can create as you wish.

## Example and Format

—
Now you need to generate a revised query based on the information below.

### Input
- **MCP Server information**:
[MCP Server Name] {mcp_name}
[MCP Server Description] {mcp_description}
- **Tool information**:
[Tool Name] {tool_name}
[Tool Description] {tool_description}
[Tool Schema] {tool_schema}
- **User Query**: {query}
- **Environmental Context**:
{environment context}

### Output

Figure 14: Prompt for slot-fill revision in Section 3.2 to supplement missing tool parameters.

**Prompt 5: WizardLM Evolution Prompt**

I want you act as a Prompt Rewriter.
Your objective is to rewrite a given prompt into a more complex version to make those famous AI systems (e.g., chatgpt and GPT4) a bit harder to handle.
But the rewritten prompt must be reasonable and must be understood and responded by humans.
Your rewriting cannot omit the non-text parts such as the table and code in #The Given Prompt#:. Also, please do not omit the input in #The Given Prompt#.
You SHOULD complicate the given prompt using the following method:
{}
You should try your best not to make the #Rewritten Prompt# become verbose, #Rewritten Prompt# can only add 10 to 20 words into #The Given Prompt#.
'#The Given Prompt#', '#Rewritten Prompt#', 'given prompt' and 'rewritten prompt' are not allowed to appear in #Rewritten Prompt#

Figure 15: Prompt for WizardLM evolution in Section 3.2 to increase query complexity and diversity.

**Prompt 6: LLM Quality Filtering Prompt**

You are an expert in information retrieval and query optimization. Your task is to evaluate the quality of the following query:

**"{query}"**.

When assessing the query, consider:
1. **Clarity** – Is the query unambiguous and easy to understand?
2. **Specificity** – Does it include enough detail to retrieve relevant results?
3. **Relevance** – Is it likely to produce results aligned with the user's intent?
4. **Completeness** – Does it provide all necessary context or constraints?

## Output Format
[Score]: 1–10 (10 = excellent)

Figure 16: Prompt for LLM-based quality filtering of generated instructions, as elaborated in Section 3.3.

**Prompt 7: Weather MCP Quality Assessment Prompt**

You are an expert evaluator. Given a user query and multiple answers from different MCPs, score each answer on a 0-5 scale.

User Query: {query}

MCP Answers:
{answers_text}

Scoring Criteria:
- 0: Answer is irrelevant, unhelpful, or "I don't know"
- 1: Answer is barely relevant but provides minimal useful information
- 2: Answer is somewhat relevant and provides basic information
- 3: Answer is relevant and provides good information
- 4: Answer is very relevant and provides detailed, useful information
- 5: Answer is excellent, comprehensive, and directly addresses the query perfectly

Respond with a JSON object mapping MCP names to scores (0-5 integers only):
{"MCP Name": score, ...}

Figure 17: Prompt for weather MCP quality assessment using a 0-5 point scale, as discuessed in Section B.1.

