# OpenReview forum: "MCP-Flow: Facilitating LLM Agents to Master Real-World, Diverse and Scaling MCP Tools"
_ICLR.cc/2026/Conference — ICLR 2026 Conference Withdrawn Submission_

### Official Review · Reviewer_qUzx · 2025-10-28

**Soundness:** 2
**Presentation:** 3
**Contribution:** 2
**Rating:** 2
**Confidence:** 4

**Summary:**

This paper introduces MCP-Flow, an automatic pipeline to collect MCP tools and construct corresponding function calling datasets for improving LLM capability when facing real-world MCP tools. The pipeline includes web-agent server crawling, tool processing, data generation (with methods including few-shot prompting and instruction evolution), data filtration. The experiments are conducted on the collected data, around 70k in total, with data split to seen set, unseen tools and unseen servers. Results reveal that models with smaller size trained with their data outperform large models, and generalize well to unseen tools and unseen servers. Moreover, the data can also serve as database for sample retrieval, improving larger models in training-free manner.

**Strengths:**

1. A new pipeline which has been proven that effectively enables MCP tool crawling and data generation.

2. The release of a new dataset containing around 70k real-world MCP tool samples, which exceeds the size of most currently available function-calling datasets.

3. Interesting findings regarding data utilization in a training-free manner.

**Weaknesses:**

1. The pipeline description appears largely engineering-oriented, and it lacks innovative design that is interesting or distinct from existing work.

2. The experiments are conducted only on the authors’ generated function calling data. Given the availability of many existing synthetic or MCP-based benchmarks (e.g., BFCL, $\tau$-bench, MCP-Universe, etc.), it is not acceptable that no results on established benchmarks are reported.

3. The dataset seems to focus mainly on single-turn instructions. While this is important, recent work has shown that multi-turn scenarios are more representative of real-world applications, so corresponding data construction and evaluation are expected.

4. The pipeline heavily relies on DeepSeek-V3 for quality checking, but it lacks analysis of agreement between human annotators and the model.

5. The pipeline simply leverages WizardLM-like evolution operations to increase complexity, while tool candidates are randomly sampled. It is therefore difficult to assess whether the resulting instructions are sufficiently complex, especially considering that similar tools can increase tool-selection difficulty.

**Questions:**

1. What is difference between metrics Param and AST? As far as I know, AST already evaluates both tool selection and parameter filling.

2. Why large models without training also perform worse on the unseen data? If I understand correctly, All test sets should be unseen for models without training.

---

### Official Review · Reviewer_4k2X · 2025-10-30

**Soundness:** 2
**Presentation:** 2
**Contribution:** 3
**Rating:** 4
**Confidence:** 4

**Summary:**

This work focuses on collecting MCP tool-calling trajectories that can be used to fine-tune LLMs. The authors propose MCP-Flow, an automated, web-agent–driven pipeline that integrates large-scale server discovery, data synthesis, and model training. MCP-Flow consists of two major components:  server discovery and data synthesis, and the data synthesis pipeline comprises two main stages: data generation and data filtration.

Experimental results demonstrate that agents fine-tuned with MCP-Flow significantly outperform baseline models in tool selection, function call accuracy, and overall task success.

**Strengths:**

- The paper tackles a highly practical problem — enabling LLMs to effectively operate within the rapidly expanding MCP ecosystem of real-world tools.

- The proposed dataset provides valuable large-scale resources for the research community, covering 1k+ MCP servers, 11k+ tools, and 68k instruction–function call pairs.

**Weaknesses:**

- Diversity metrics rely on embedding cosine distances and LLM-based judge scores, but no human annotator agreement is reported.
Incorporating even a small-scale human evaluation would strengthen the claim that the synthetic data aligns with human quality judgments.

- The paper does not sufficiently discuss how the dataset handles multi-turn interactions, which are essential for realistic agentic tasks.

- The introduction section could be improved for smoother narrative flow — some transitions between motivation and methodology are slightly disjointed.

**Questions:**

- Is there any human verification or manual validation involved during Server Discovery stage?

---

### Official Review · Reviewer_xeKy · 2025-11-01

**Soundness:** 3
**Presentation:** 3
**Contribution:** 3
**Rating:** 4
**Confidence:** 3

**Summary:**

This paper addresses limitations in LLMs’ utilization of the MCP ecosystem, small server coverage, costly manual curation, and lack of training support. It introduces MCP-Flow, an automated web-agent-driven pipeline for large-scale MCP server discovery, data synthesis, and model training. MCP-Flow collects data from 1,166 servers and 11,536 tools across 6 marketplaces, generating 68,733 high-quality instruction-function call pairs and 6,439 trajectories. It includes two key components: web-agent-based server crawling (with deduplication and local deployment) and scalable data synthesis (few-shot generation, slot-fill revision, WizardLM evolution, and rigorous filtration). Experiments show MCP-Flow outperforms SOTA LLMs (e.g., GPT-4o, Claude-4-Sonnet) in MCP tool selection/formatting (even with small models like Qwen3-0.6B), enhances closed models via retrieval augmentation, and improves agent performance on the GAIA benchmark while reducing costs.

**Strengths:**

1. This paper constructs a large-scale MCP dataset with broad coverage, including 1,166 servers, 11,536 tools, 68,733 instruction-function call pairs, and 6,439 trajectories—far exceeding the scale of existing MCP-related works and truly reflecting the complexity of the real-world MCP ecosystem.
2. This paper proposes a web-agent-driven automated pipeline for server discovery and data synthesis, eliminating costly manual curation. The pipeline supports incremental updates for new servers, adapting to the rapid evolution of the MCP ecosystem while reducing time and computational costs.

**Weaknesses:**

1. Limited analysis of tool quality variance and standardized MCP server evaluation.
2. More OOD evaluation is needed, eg. MLE-Bench.

**Questions:**

As shown in the Weakness.

---

### Official Review · Reviewer_wn4G · 2025-11-01

**Soundness:** 2
**Presentation:** 3
**Contribution:** 2
**Rating:** 4
**Confidence:** 4

**Summary:**

This paper presents a data synthesis framework for multiple MCP tools. The authors collect a variety of real MCP tools from the official MCP server and construct a tool graph to represent their relationships. By sampling candidate tools from this graph, the method leverages an LLM to synthesize instruction–response trajectories. A combination of rule-based and LLM-based filtering is then applied to refine the generated data. The final dataset is released together with the code, aiming to facilitate further research in multi-tool data synthesis.

**Strengths:**

1. The paper is well-written and clearly organized, with professional presentation and figures.
2. The release of code and data is commendable, as it provides valuable resources for the community and may stimulate further research in MCP-related data generation and tool-use modeling.

**Weaknesses:**

1. The contribution appears somewhat incremental, resembling an extension of ToolACE that integrates the MCP server into a similar data synthesis pipeline. The paper would benefit from a clearer articulation of its unique insights or methodological advances beyond existing frameworks.

2. In Section 3.1, the authors describe the MCP tool collection process. However, MCP tools are continuously updated or deprecated on the server. The paper does not discuss how such versioning and temporal drift are handled during data collection and maintenance, which raises concerns about dataset reproducibility.

3. During instruction synthesis, the method focuses solely on problems that explicitly require tool usage. Real-world interactions, however, often involve queries solvable without tools. Ignoring such cases may bias the dataset and limit the model's ability to decide when not to call a tool.

4. The current synthesis process appears restricted to single-step tool calls. It does not consider multi-round dependencies, such as inter-tool data flow or user clarification when parameters are missing. These aspects are essential for building realistic multi-turn tool-use data.

5. Experiments are only conducted on the self-synthesized dataset. To validate generalization and robustness, the model trained on this dataset should be evaluated on existing public tool-use benchmarks, demonstrating transferability to unseen tools and domains.

6. The paper would be improved by analyzing the domain coverage of the synthesized dataset (e.g., tool types, functional categories).

7. No ablation studies are provided to quantify the impact of important components such as the tool-graph construction and filtering strategies. A systematic analysis of these steps would strengthen the empirical evidence for the proposed design choices.

**Questions:**

Please refer to the weaknesses

---

### Note · Authors · 2026-01-04

I have read and agree with the venue's withdrawal policy on behalf of myself and my co-authors.